# Impact of Serotonin Transporter Absence on Brain Insulin Receptor Expression, Plasma Metabolome Changes, and ADHD-like Behavior in Mice fed a Western Diet

**DOI:** 10.3390/biom14080884

**Published:** 2024-07-23

**Authors:** Daniel C. Anthony, Fay Probert, Anna Gorlova, Jenna Hebert, Daniel Radford-Smith, Zlata Nefedova, Aleksei Umriukhin, Andrey Nedorubov, Raymond Cespuglio, Boris Shulgin, Aleksey Lyundup, Klaus Peter Lesch, Tatyana Strekalova

**Affiliations:** 1Department of Pharmacology, Oxford University, Oxford OX1 3QT, UK; daniel.anthony@pharm.ox.ac.uk (D.C.A.); fay.probert@chem.ox.ac.uk (F.P.); hebertjcr@gmail.com (J.H.); daniel.radford-smith@some.ox.ac.uk (D.R.-S.); 2Department of Chemistry, Oxford University, Oxford OX1 2JD, UK; 3Research and Education Resource Center, Peoples Friendship University of Russia (RUDN University), 117198 Moscow, Russia; anna.gorlova204@gmail.com (A.G.); raymond.cespuglio@sfr.fr (R.C.); lyundup2020@gmail.com (A.L.); 4Department of Normal Physiology, Sechenov First Moscow State Medical University, 119991 Moscow, Russia; zlanefedova@gmail.com (Z.N.); alum1@yandex.ru (A.U.); nedorubov.ras@gmail.com (A.N.); 5Laboratory of Engineering Profile Physical and Chemical Methods of Analysis, Korkyt Ata Kyzylorda University, Kyzylorda 120014, Kazakhstan; bshulgin83@gmail.com; 6Endocrinology Research Centre, Dmitry Ulyanov Str. 19, 117036 Moscow, Russia; 7Division of Molecular Psychiatry, Center of Mental Health, University Hospital Würzburg, 97080 Würzburg, Germany; kplesch@mail.uni-wuerzburg.de; 8Department of Child and Adolescent Psychiatry, Psychosomatics and Psychotherapy, Center of Mental Health, University Hospital Würzburg, 97080 Würzburg, Germany

**Keywords:** serotonin transporter, *Sert*-deficient mice, western diet, metabolomics, glucose tolerance, insulin receptors, ADHD, aging

## Abstract

The impaired function of the serotonin transporter (SERT) in humans has been linked to a higher risk of obesity and type 2 diabetes, especially as people age. Consuming a “Western diet” (WD), which is high in saturated fats, cholesterol, and sugars, can induce metabolic syndrome. Previous research indicated that mice carrying a targeted inactivation of the *Sert* gene (knockout, KO) and fed a WD display significant metabolic disturbances and behaviors reminiscent of ADHD. These abnormalities might be mediated via a dysfunction in insulin receptor (IR) signaling, which is also associated with adult ADHD. However, the impact of *Sert* deficiency on IR signaling and systemic metabolic changes has not been thoroughly explored. In this study, we conducted a detailed analysis of locomotor behavior in wild-type (WT) and KO mice fed a WD or control diet. We investigated changes in the blood metabolome and examined, via PCR, the expression of insulin receptor A and B isoforms and key regulators of their function in the brain. Twelve-month-old KO mice and their WT littermates were fed a WD for three weeks. Nuclear magnetic resonance spectroscopy analysis of plasma samples showed that KO mice on a WD had higher levels of lipids and lipoproteins and lower levels of glucose, lactate, alanine, valine, and isoleucine compared to other groups. SERT-KO mice on the control diet exhibited increased brain levels of both IR A and B isoforms, accompanied by a modest increase in the negative regulator ENPP. The KO mice also displayed anxiety-like behavior and reduced exploratory activity in an open field test. However, when the KO animals were fed a WD, the aberrant expression levels of IR isoforms in the KO mice and locomotor behavior were ameliorated indicating a complex interaction between genetic and dietary factors that might contribute to ADHD-like symptoms. Overall, our findings suggest that the lack of *Sert* leads to a unique metabolic phenotype in aged mice, characterized by dysregulated IR-related pathways. These changes are exacerbated by WD in the blood metabolome and are associated with behavioral abnormalities.

## 1. Introduction

Metabolic syndrome increases the risk of type 2 diabetes and other serious somatic pathologies, raising mortality rates approximately 1.5-fold [1,2]. It has also been linked to a more severe clinical course and increased risk of mortality from COVID-19 [3]. The global prevalence of metabolic syndrome is rising [4], with a higher incidence in women than men, and a nearly fivefold increase among older adults [5,6]. The development of metabolic syndrome may result from interactions between lifestyle and genetic factors [7]. A deficiency of the serotonin transporter (SERT) is one genetic factor associated with metabolic disorders [8]; however, the mechanisms underlying this relationship are not fully understood.

SERT is a key component of serotonergic neurotransmission [9,10], regulating extracellular serotonin levels and playing a role in metabolic processes [11,12,13,14,15]. A genetic variant of gene encoding SERT (*Slc6a4*) that reduces its expression, known as the “short” (S) allele, is associated with a higher body mass index (BMI) [16,17,18] and the development of type 2 diabetes [19]. This variant is present in 20–60% of the human population. *Sert* knockout (KO) mice and rats, which are widely used models of decreased SERT function, demonstrate compromised glucose tolerance, a late-onset obesity phenotype, and increased scores of anxiety-like and depressive-like behaviors [20,21,22,23,24,25].

The Western diet (WD), characterized by high contents of saturated fats, cholesterol, and simple sugars, is a leading cause of obesity and related metabolic dysfunctions [26,27,28]. Previous studies using a model of three-week-long WD feeding in mice [29,30,31,32,33] revealed that in aged female mice, the genetic absence of *Sert* exacerbates the effects of WD on glucose metabolism, body weight gain, behavioral parameters of emotionality, and the expression of inflammation markers [23,34]. Specifically, in female SERT-KO mice, compared to wild-type (WT) mice, WD led to more significant decreases in glucose tolerance, overexpression of Toll-like receptor 4 in the brain, behavioral abnormalities such as disrupted hippocampus-dependent performance, depressive-like behavior, and decreased novelty exploration. Insulin resistance, obesity, hepatosteathosis have all been reported previously to be a feature in male SERT-KO mice, irrespective of dietary regimen [20].

Given the combination of metabolic and pro-inflammatory changes produced by WD in SERT-KO mice, we aimed to further investigate these aspects in our model. Accumulating evidence suggests that KO mice exposed to WD are prone to developing neuropsychiatric symptoms, particularly ASD-like and ADHD-like behaviors, such as aberrations in social and locomotor behaviors, cognitive and emotional abnormalities [23,24,33,35,36]. Decreased SERT function was shown to be inhibitory of the IRS1/PI3K/Akt signaling, which is a key pathway in insulin signal transduction [14,15,37,38]. The metabolic consequences of WD consumption and the ADHD-like syndrome have been found to involve altered insulin receptor (IR) function [39,40,41,42], that seem to be underpinned by variation on the expression and function of the A and B isoforms of the receptor [15,43,44]. In the present study, we also examined leptin signaling, which plays a pivotal role in the regulation of energy homeostasis [45,46] and is linked to insulin regulation [47,48]. The impairment of central leptin and insulin-related pathways have been suggested to underlie diet-induced behavioral abnormalities [48]. Hence, we sought to study the expression of IR and leptin receptor isoforms and changes in IR-related pathways, in addition to examination of the systemic changes in the metabolome, which have, hitherto, not been studied in SERT-KO mice fed WD.

To address these questions, we sought to characterize the plasma metabolome of mice with genetic SERT absence under both basal conditions and after WD exposure. Metabolomics, the downstream summation of genetic and environmental factors, has been widely applied for diagnosing metabolic diseases, discovering biomarkers, and exploring underlying disease mechanisms [49]. The high clinical significance of metabolomics has been demonstrated, for example, by distinguishing between stages of multiple sclerosis [50,51] and types of ulcerative colitis [52] using patient blood samples. In metabolomics, the relative levels of metabolites, including lipids, amino acids, and carbohydrates, measured via global metabolite profiling [53], reflect the influence of both genetic and environmental factors, including diet [54]. In animal models, metabolome profiling has been used to characterize phenotypic changes induced by stress [55,56], pro-inflammatory challenges [57], and diet [58]. Specifically, Guo et al. [58]), using ^1^H-NMR-based metabolomics in a model of high-fat and high-cholesterol dietary exposure in hamsters, showed metabolite alterations related to energy homeostasis, intestinal microbiota functions, inflammation, and oxidative stress. To date, changes in plasma metabolites induced by SERT deficiency have not been explored, despite its involvement in the development of metabolic diseases.

Additionally, we also investigated the gene expression of phosphatase and tensin homolog (PTEN), a major regulator of IR [59], indicating potential reduction in glucose consumption by neurons and neuroglia and the prevalence of ketone metabolism [60,61]. We also examined the expression of acyl-CoA synthetase-1 (ACSL1), which is crucial for triacylglycerol (TAG) synthesis, fatty acid β-oxidation, and the enhancement of inflammatory responses via toll-like receptors (TLRs), TNF-α, IFN-γ, and pro-inflammatory interleukins (e.g., IL-1β, IL-18) [62], whose overexpression was shown in WD-fed mice [30]. The expression of CD36, a scavenger receptor involved in tissue uptake of fatty acids and intracellular lipid storage, was also studied. Excessive CD36 expression under a high-fat diet is associated with metabolic dysfunction [63], pro-inflammatory responses [64], and insulin resistance [65]. Patients with type 2 diabetes mellitus display elevated levels of tyrosine phosphatase 1B (PTPN1), and membrane IR-associated glycoprotein PC-1 (ENPP1), which downregulate IR phosphorylation and insulin signal transduction [66,67,68,69].

In the current study, we addressed the blood metabolic profile and insulin and leptin signaling in the brains of aged female mice, including homozygous *Sert*-KOs and their WT littermates, fed either a control diet or WD. We chose to use female mice as female rodents are documented to be more susceptible to the effects of the WD than males [11,29,70] and our previous experiments extensively validated the WD paradigm in female mice [23,24,30,31,33]. Plasma samples were analyzed via nuclear magnetic resonance spectroscopy followed by orthogonal partial-least squares discriminant analysis and metabolite profiling. Changes in behavioral parameters of locomotion, anxiety, and exploration were studied using the open field test. We examined the gene expression of IR isoforms A and B, leptin receptor isoforms A and B, and functionally related molecules: ACSL1, ENPP, PTEN, CD36, and PTPN1.

## 2. Materials and Methods

### 2.1. Animals

Experiments, in two cohorts (A & B), were performed using 12-month-old homozygous female SERT KO mice and WT littermates born from heterozygous mutants at the tenth generation (F10) of backcrosses with WT C57BL/6J mice (Figure 1A,B). Mice were housed 3–4 per cage during the study, under a reversed 12 h light-dark cycle (lights on: 21:00) with food and water *ad libitum* and under controllable laboratory conditions (22  ±  1 °C, 55% humidity). Laboratory housing conditions and experimental procedures were set up and maintained in accordance with the European Communities Council Directive for the care and use of laboratory animals (2010/63/EU) and approved by the local ethics committee of Oxford University (PPL number: P996B4A4E, Granted: 7 December 2018). All experiments were compliant with ARRIVE guidelines (http://www.nc3rs.org.uk/arrive-guidelines, accessed on 2 February 2024). 

### 2.2. Study Flow and Dietary Challenge

Mice were fed with a standard laboratory diet (control diet, CD; groups WT/CD and KO/CD) with an energy content of 3.8  kcal/g, 4.3% of fat (1.3 of saturated fat) (D18071801, Research Diet Inc., New Brunswick, NJ, USA) or with a diet containing 0.2% cholesterol, 21.3% of fat (10.5% of saturated fat), and an energy content of 4.6  kcal/g, Western diet (WD, D11012302, Research Diet Inc., New Brunswick, NJ, USA; groups WT/WD and KO/WD) for three weeks as described elsewhere [33]. The content of the nutrients in calories, weight, and the ingredients are indicated in Appendix A. Body weight was monitored weekly. After a three-week period of dietary intervention, one cohort of mice was subject to the glucose tolerance test followed by blood collection for the analysis of leptin content (Figure 1A). Another cohort of animals were exposed to the same dietary conditions, tested in the open field test, and culled for blood collection for metabolomics analysis; the brain was removed and dissected for gene expression analysis (Figure 1B). A total of 6 to 8 mice per group were used in each experiment. In the first experiment, 28 mice were used (7 mice in each group, Figure 1A), in the second cohort, 31 mice were used (Figure 1B), i.e., a total of 59 mice.

#### 2.2.1. Open Field Test

The open field test was carried out during an active period of the animals’ light cycle (09:00–21:00) as previously described [71]. The open field apparatus consisted of a square arena (40 cm × 40 cm). Mice were put in the center of the open field arena, and their behavior was video recorded for 5 min under 25 Lux lighting. Behavior was analyzed off-line manually and using the EthoVision software (XP 12.5, NoldusEthoVision BV, Wageningen, The Netherlands). Mean velocity, number of transitions from central (square 20 cm × 20 cm) zone to peripheral zone and back, time spent with stretched posture, and the duration of grooming were all recorded. Data were presented in absolute values for all the groups and were normalized to the respective genotype control diet group in WD-fed groups.

#### 2.2.2. Glucose Tolerance Test

The oral glucose tolerance test (OGTT) was performed as described by Veniaminova et al. [23]. Mice were fasted overnight for 18 h, starting at 16:00. Following the fasting period, a glucose solution (2 g/kg, 1.8 g/L) was administered via oral gavage. Blood samples were collected from the tail vein at the following time points: prior to glucose administration (0 min) and at 5, 15, and 30 min after administration. Blood glucose levels were measured using the OneTouch UltraEasy glucometer and strips (LifeScan OneTouch, Dubai, United Arab Emirates). The area under the curve (AUC) for the test period was calculated and normalized to basal glucose levels to assess glucose tolerance.

#### 2.2.3. Tissue Dissection and Blood Collection

One cohort of mice was euthanized using isoflurane inhalation, and blood samples for leptin level analysis were collected via cardiac puncture. The remaining mice were terminally anesthetized via isoflurane inhalation. Blood samples for metabolomics analysis were also collected via cardiac puncture. Following blood collection, each mouse’s brain was perfused with saline and subsequently dissected as previously described [30]. The hippocampus, dorsal raphe, prefrontal cortex, and hypothalamus were isolated and stored at −80 °C until use.

#### 2.2.4. RNA Extraction and RT-qPCR

mRNA was extracted using the RNeasy Mini Kit (Qiagen, Venlo, The Netherlands) as previously described [30]. First-strand cDNA synthesis was performed using the High-Capacity cDNA Reverse Transcription Kit (Applied Biosystems, Waltham, MA, USA); 1 μg total RNA was converted into cDNA. Quantitative PCR for the genes of interest (insulin receptors isoforms A (IRA) and B (IRB), ACSL1, ENPP, PTPN1, PTEN, CD36) and the reference genes (glyceraldehyde 3-phosphate dehydrogenase (Gapdh), beta-actin (Actb), beta-2 microglobulin (B2m)) was performed using the SYBR Green PCR Master Mix (Applied Biosystems, Applied Biosystems, Waltham, MA, USA) and QuantStudio 7 Flex Real-Time qPCR System (Applied Biosystems, Applied Biosystems, Waltham, MA, USA). Sequences of primers used are presented in Appendix A. Reference genes for normalization were tested for stability using RefFinder software. Results (https://www.ciidirsinaloa.com.mx/RefFinder-master/, accessed on 2 February 2024) of RT-qPCR measurement were expressed as Ct values, and the comparative Ct method [72] was used. Data are given as expression folds compared to the mean expression values in WT/CD.

#### 2.2.5. Blood Biochemical Analysis of Leptin Concentration

Blood was centrifuged at 10,000× *g* for 10 min at 4 °C in in heparinized tubes. Serum was collected and stored at −20 °C until later use. A commercially available Mouse Leptin (OB) ELISA Kit (Sigma-Aldrich, St. Louis, MA, USA) was used to measure leptin levels; the optical densities of experimental plates were measured at 450 nm using a plate reader (Wallac 1420 VICTOR, Waltham, MA, USA). All samples were run in duplicate. All procedures were performed according to the instruction manual and carried out as previously described [33].

#### 2.2.6. Nuclear Magnetic Resonance (NMR) Spectroscopy 

H nuclear magnetic resonance (NMR) spectroscopy, processing, and analysis were performed as previously described [53]. Briefly, 150 µL plasma was mixed with 400 µL 75 mM sodium phosphate D_2_O buffer (pH = 7.4). Carr-Purcell-Meiboom-Gill (CPMG) spectra were acquired with a 700-MHz Bruker AVII spectrometer (*Bruker* Daltonics GmbH & Co, Bremen, Germany Spectra were phased, baseline-corrected (using a 3rd degree polynomial) and chemical shifts referenced to the CH_3_-lactate doublet at δ = 1.33 parts per million (ppm) in Topspin 4.0.5 (*Bruker* Daltonics GmbH & Co Bremen, Germany. They were exported to ACD/Labs Spectrus Processor Academic Edition 12.01 (Advanced Chemistry Development, Inc., Toronto, ON, Canada) and divided into 0.02 ppm “bins”. The integral of each bin was calculated, normalized to the sum of all bin integrals in the spectrum, and Pareto-scaled.

For statistical analysis, bin integrals were exported to R software (R Foundation for Statistical Computing, https://www.r-project.org/, accessed on 2 February 2024) for principal component analysis (PCA) and orthogonal partial least squares discriminant analysis (OPLS-DA) using the ropls package [73]. Principal component analysis (PCA) was used to provide a preliminary look at whether groups separated into clusters using an unsupervised approach. We then used OPLS-DA to determine if metabolic profiles were significantly different between groups and, if so, which metabolites were responsible for the variation. 

OPLS-DA models were built from multiple iterations of randomly separated data into training sets and test sets. Training sets were used to build the models using a seven-fold internal cross-validation procedure. The models were then tested on the independent test data in an eight-fold external cross-validation scheme to determine the accuracy, sensitivity, and specificity. A total of 800 models were built for each pairwise group comparison. In addition, the same external cross-validation with repetition procedure was used to build models from data randomly sorted into classes (permutation testing); the average accuracy of these models was always approximately 50%. If the true models performed significantly better than random chance in categorizing the samples in the test sets (*p* < 0.001, Kolmogorov–Smirnov test), the models were considered significant. For significant models, the metabolites driving the group separations were identified using the average variable importance in projection (VIP) scores. Metabolites corresponding to bins with VIP scores greater than 1.5 in at least one of the four pairwise comparisons were identified via the reference literature values [74,75] and 2D total correlation spectroscopy (TOCSY) spectra. Integral values for each metabolite region were normalized to the WT/CD control group and plotted in a heatmap and boxplots to compare metabolite levels between all four groups.

#### 2.2.7. Statistics

NMR spectroscopy data were analyzed as described above. All other data were analyzed using GraphPad Prism version 8.01 (San Diego, CA, USA). All quantitative data sets were first analyzed for normal distribution using the Shapiro–Wilk normality test, then, as all the data were distributed normally, two-way ANOVA followed by Tukey’s multiple comparisons test. *T*-test or one sample *t*-test were used for normalized data. The level of significance was set at *p* < 0.05. Data are presented as boxplots or mean ± SEM.

## 3. Results 

### 3.1. Effects of WD and SERT Deficiency on Metabolism 

We found a significant genotype effect on body weight at week 1 and week 2 (F = 12.36, *p* = 0.0019, and F = 17.29, *p* = 0.0004, respectively, two-way ANOVA, Figure 2A,B). Body weight was increased in KO mice compared to WT animals. At week 3, both diet and genotype had significant effects on body weight (F = 7.261, *p* = 0.0132, and F = 14.8, *p* = 0.0009, respectively, Figure 2C). KO mice had greater body weight than WT mice, and mice fed a Western diet (WD) had greater body weight than those fed a CD. Normalization to genotype revealed a significant increase in body weight of KO mice fed a WD at week 3 compared to 100% (*p* = 0.0075, one-sample *t*-test, Figure 2C). We observed a significant diet effect (F = 9.207, *p* = 0.0084, two-way ANOVA, Figure 2D) on the area under the curve (AUC) for glucose tolerance at three weeks. Mice fed a WD had increased AUC compared to those fed a CD. Significant effects of diet (F = 10.08, *p* = 0.0088) and genotype (F = 19.50, *p* = 0.0010) were also found on leptin serum levels (two-way ANOVA, Figure 2E) at week 3. KO mice had higher leptin levels than WT mice, and leptin levels were increased in WD-fed mice compared to CD-fed groups. Normalization to genotype showed that a WD led to a significant increase in the AUC for glucose tolerance in WT animals (*p* = 0.0279, one-sample *t*-test, Figure 2D). Both WT and KO mice fed a WD showed increased leptin levels compared to 100% (*p* = 0.0449 and *p* = 0.0087, respectively). Additionally, leptin levels were higher in WT animals (*p* = 0.0013, *t*-test, Figure 2E). All statistical values are presented in Appendix A.

### 3.2. SERT-KO Genotype, WD, and the Combination of Genotype and Diet Produce Distinct Plasma Metabolic Profiles

PCA was used for unsupervised exploratory analysis to investigate how genotype and Western diet alter plasma metabolites. Average ^1^H CPMG spectra of WT/CD, WT/WD, KO/CD, and KO/WD are shown in Appendix A. The first principal component (PC1) of the PCA plot (Figure 3A) accounted for 40.8% of the variance in the data. PC1 separated the control group (WT/CD) and the KO/WD mice, but the WT/WD and KO/CD overlapped with these groups. This result was suggestive of an interaction between genotype and diet and that the combination of SERT deficiency and the WD resulted in more robust changes in metabolic profiles compared to the KO genotype or WD alone. Boxplots showing the average percent change for the most important metabolites contributing to the group separations relative to the WT/CD group are presented in Appendix A.

To further explore metabolic differences in more detail, OPLS-DA was used. Four pairwise group comparisons were performed. These comparisons and a summary of the OPLS-DA models are outlined in Figure 3B. The average performance of the OPLS-DA models for all four comparisons was significantly better than chance (Figure 3B). In both WT and KO mice, the WD resulted in significantly different metabolic profiles compared to the CD. In addition, the absence of *Sert* gave rise to a distinct metabolic profile compared to WT animals. This difference was detected via OPLS-DA regardless of whether the mice were fed a CD or WD (Figure 3B).

### 3.3. Sert Deficiency and WD Have Additive Effects on Lipid, Lactate, Alanine, and Valine Levels

The OPLS-DA results confirmed that genotype, diet, and their combination explained the variation in plasma metabolites. We then used VIP scores to identify which spectral bins were crucial for constructing the OPLS-DA models. These bins, along with the VIP scores and metabolite assignments, are summarized in Appendix A. Comparisons of metabolite levels across all groups are shown in the boxplots and a heatmap (Figure 3B,C). The largest changes in lipids, lactate, alanine, and the branched-chain amino acid (BCAA) valine, compared to the WT/CD group, were found in the KO/WD animals. Specifically, a significant genotype effect was found for lipid level (F = 13.61, *p* = 0.0014, two-way ANOVA); it was higher in KO mice compared to WT (Appendix A). A significant genotype x diet interaction was present for unsaturated lipid concentration (F = 4.339, *p* = 0.0491) that was elevated in the KO/WD group in comparison with WT/WD animals (*p* = 0.0466, Tukey’s test; Appendix A). Significant genotype and diet effects were shown for VLDL level (F = 4.906, *p* = 0.0379 and F = 8.572, *p* = 0.008, respectively; Appendix A). 

In contrast, lactate levels were reduced in both the KO/CD and WT/WD groups, with the largest reduction observed in the KO/WD animals. Similarly, alanine and valine followed patterns similar to that of lactate. Specifically, an effect of diet was significant for lactate concentration (F = 16.05, *p* = 0.0006; Appendix A). Significant genotype and diet effects were found for alanine concentration (F = 25.95, *p* < 0.0001 and F = 24.63, *p* < 0.0001, respectively; Appendix A), isoleucine concentration (F = 8.356, *p* = 0.0085 and F = 27.56, *p* < 0.0001, respectively; Appendix A), and valine concentration (F = 20.69, *p* = 0.0002 and F = 17.96, *p* = 0.0004, respectively; Appendix A). A genotype x diet interaction well as genotype effect were significant for glucose (F = 9.263, *p* = 0.0062 and F = 16.38, *p* = 0.0006). This measure was significantly lower in KO/WD group than in the WT/WD group (*p* = 0.0001; Appendix A). In these cases, the KO genotype and WD had additive effects on metabolite levels. The OPLS-DA results confirmed that genotype, diet, and the combination of genotype and diet explained variation in plasma metabolites. All statistical values are presented in Supplementary Appendix A.

### 3.4. WD and Sert Deficiency Affected Behavioral Parameters in Open Field Test

Two-way ANOVA revealed no differences between the groups in mean velocity. However, when normalized to their respective CD groups, velocity was decreased in WT/WD and increased in KO/WD (*p* = 0.0359 and *p* = 0.0264, respectively (one sample *t*-test, Figure 4A). This measure was significantly greater in the normalized KO/WD group than in normalized WT/WD mice (*p* = 0.0038, *t*-test). Significant genotype effects were found in the number of transitions between central zone and periphery (F = 7.153, *p* = 0.0130, two-way ANOVA, Figure 4B), which was decreased in KO mice. No differences were observed in time spent with a ‘stretched posture’ (Figure 4*C*). However, when normalized to CD, duration of the stretched posture was decreased in WT/WD compared to 100% (*p* < 0.0001, one sample *t*-test). The interaction between diet and genotype was significant in duration of grooming (F = 4.517, *p* = 0.0436, two-way ANOVA, Figure 4D), while this parameter when normalized to CD was increased in WT/CD compared to 100% (*p* = 0.0213, one sample *t*-test). This measure was significantly lower in the normalized KO/WD group than in WT/WD mice (*p* = 0.0034, *t*-test). All statistical values are presented in Supplementary Appendix A.

### 3.5. WD and Sert Deficiency Altered Brain Expression of Insulin Receptors 

A significant interaction between genotype and diet was observed in mRNA levels of IR isoform A (IRA) in the hippocampus (F = 12.11, *p* = 0.0019, two-way ANOVA, Figure 5A), dorsal raphe (F = 5.008, *p* = 0.0048, Figure 5B), and hypothalamus (F = 6.025, *p* = 0.0214, Figure 5D). IRA expression levels were increased in the hippocampus, dorsal raphe, and hypothalamus of KO/CD mice compared to WT/CD mice (*p* = 0.0015, *p* = 0.0071, and *p* = 0.0047, respectively, Tukey’s test). Additionally, post-hoc analysis revealed a significant decrease in IRA expression levels in the hippocampus, dorsal raphe, and hypothalamus of KO/WD mice compared to KO/CD mice (*p* = 0.0024, *p* = 0.0010, and *p* = 0.0102, respectively).

The effects of genotype and diet were significant in the prefrontal cortex (F = 12.18, *p* = 0.0019, and F = 5.229, *p* = 0.0313, respectively, Figure 5C). IRA levels were increased in KO mice compared to WT mice and decreased in WD-fed mice compared to CD-fed mice. When normalized to their respective CD groups and compared to 100%, IRA expression levels were decreased in the hippocampus, dorsal raphe, prefrontal cortex, and hypothalamus of KO/WD mice (*p* = 0.0004, *p* < 0.0001, *p* = 0.0334, and *p* = 0.0021, respectively, one-sample *t*-test, Figure 4A–D). This measure was also significantly decreased in the prefrontal cortex of WT/WD mice (*p* = 0.003). IRA expression was lower in normalized KO/WD mice compared to WT/WD mice in the hippocampus, dorsal raphe, and hypothalamus (*p* = 0.0006, *p* = 0.0098, and *p* = 0.0042, respectively, *t*-test).

A significant interaction between genotype and diet was also observed in mRNA levels of IR isoform B (IRB) in the hippocampus (F = 12.11, *p* = 0.0019, two-way ANOVA, Figure 5E) and hypothalamus (F = 6.025, *p* = 0.0214, Figure 5H). IRB expression was significantly decreased in the hippocampus and hypothalamus of KO/WD mice compared to KO/CD mice (*p* = 0.0475 and *p* = 0.0003, respectively, Tukey’s test). In the hypothalamus, IRB levels in KO/CD mice were significantly higher than in WT/CD mice (*p* = 0.0052). The effect of genotype was significant in the dorsal raphe and prefrontal cortex (F = 15.99, *p* = 0.0007, and F = 12.63, *p* = 0.0016, respectively, Figure 5F,G). IRB levels were increased in KO mice compared to WT mice in the dorsal raphe and prefrontal cortex. When normalized to their respective CD groups and compared to 100%, IRB expression levels were similarly decreased in the hippocampus, dorsal raphe, prefrontal cortex, and hypothalamus of KO/WD mice (*p* = 0.0005, *p* = 0.0429, *p* = 0.0497, and *p* < 0.0001, respectively, one-sample *t*-test, Figure 5E–H). IRB expression was lower in normalized KO/WD mice compared to WT/WD mice in the hippocampus, dorsal raphe, and hypothalamus (*p* = 0.0022, *p* = 0.0446, and *p* = 0.0012, respectively, *t*-test). All statistical values are presented in Supplementary Appendix A.

### 3.6. WD and Sert Deficiency Changed Brain Expression of Transcription and Signaling Factors

A significant interaction between genotype and diet was observed for ACSL1 expression in the hippocampus (F = 7.709, *p* = 0.012, two-way ANOVA, Figure 6A). ACSL1 expression was lower in KO/WD mice compared to WT/WD mice (*p* = 0.0204, Tukey’s test). When normalized to their respective CD groups and compared to 100%, ACSL1 expression in the hippocampus was increased in KO mice (*p* = 0.0293, one-sample *t*-test) and was significantly higher than in the normalized WT group (*p* = 0.0031, *t*-test). A significant diet effect was revealed for ACSL1 expression in the dorsal raphe (F = 4.978, *p* = 0.0379, Figure 6B), which was higher in mice housed on WD. ACSL1 expression was also increased in KO mice normalized to their respective CD group and compared to 100% (*p* = 0.0278, one-sample *t*-test). However, two-way ANOVA revealed no differences between the groups in ACSL1 expression in the prefrontal cortex and hypothalamus (Figure 6C,D).

A significant genotype effect was observed for ENPP expression in the hippocampus (F = 13.42, *p* = 0.0017, two-way ANOVA), with decreased levels in KO/CD mice compared to WT/CD mice (*p* = 0.0046, Tukey’s test, Figure 6E). ENPP expression was also decreased in KO animals when normalized to their respective CD group and compared to 100% (*p* = 0.0064, one-sample *t*-test). No significant differences between groups were found for ENPP expression in the dorsal raphe, prefrontal cortex, and hypothalamus (Figure 6F–H). However, when normalized to their respective CD group and compared to 100%, ENPP expression in the dorsal raphe was increased in KO/WD mice (*p* = 0.0345, one-sample *t*-test) and was significantly higher than in the normalized WT group (*p* = 0.019, *t*-test, Figure 6F).

A significant interaction between genotype and diet was found for PTPN1 expression in the hippocampus (F = 11.88, *p* = 0.0027, two-way ANOVA, Figure 6I). PTPN1 expression was lower in both WT/CD and KO/WD mice compared to WT/WD animals (*p* = 0.016 and *p* = 0.0059, respectively, Tukey’s test). When normalized to their respective CD groups and compared to 100%, PTPN1 expression in the hippocampus was increased in KO mice (*p* = 0.0074, one-sample *t*-test) and was significantly higher than in the normalized WT group (*p* = 0.0006, *t*-test). A significant genotype effect on PTPN1 expression was observed in the dorsal raphe (F = 9.489, *p* = 0.0062, Figure 6J), with lower levels in KO mice compared to WT animals. No significant differences between groups were found in PTPN1 expression in the prefrontal cortex (Figure 6K). In the hypothalamus, a significant diet effect was revealed (F = 5.837, *p* = 0.0259, Figure 6L), with higher PTPN1 expression in groups housed on WD. When normalized to their respective CD group and compared to 100%, PTPN1 expression was significantly increased in WT/WD mice (*p* = 0.0481, one-sample *t*-test).

A significant diet effect was found for PTEN expression in the hippocampus (F = 8.608, *p* = 0.0085, two-way ANOVA, Figure 6M), with higher levels in animals housed on WD. When WT/WD mice were normalized to their respective CD group and compared to 100%, PTEN expression was significantly increased (*p* = 0.019, one-sample *t*-test). Significant genotype effects on PTEN expression were observed in the dorsal raphe and prefrontal cortex (F = 4.693, *p* = 0.0432, and F = 7.579, *p* = 0.0127, respectively, Figure 6N,O), with increased levels in the dorsal raphe of KO mice and decreased levels in the prefrontal cortex of KO animals. When normalized to their respective CD groups and compared to 100%, PTEN expression in the prefrontal cortex was lower in both WT and KO mice (*p* = 0.0111 and *p* = 0.0236, respectively, one-sample *t*-test). No significant differences between groups were found in PTEN expression in the hypothalamus (Figure 6P).

No significant differences between groups were found in CD36 expression in the hippocampus, dorsal raphe, prefrontal cortex, and hypothalamus (Figure 6Q–T). However, in the hippocampus, CD36 expression was higher in the normalized WT group compared to 100% (*p* = 0.0293, one-sample *t*-test, Figure 6Q). All statistical values are presented in Supplementary Appendix A.

## 4. Discussion

In keeping with previous reports, our study demonstrated that aged female SERT-KO mice exhibit impaired glucose tolerance and increased body weight gain compared to WT controls [23,74]. Using NMR spectroscopy followed by metabolite profiling, we investigated changes in plasma metabolites associated with Sert deficiency and the combined effects of WD and Sert knockout in mice. We observed increased levels of lipids and lipoproteins in WD-fed KO mice relative to other groups, indicating additive effects of WD and Sert deficiency on the plasma metabolome. Remarkably, there were opposing changes in velocity, duration of grooming and stretching behavior in WT vs. KO mice fed WD with respect to CD-fed groups.

The most significant metabolic changes were seen in KO mice fed a WD, which showed elevated levels of lipids and lipoproteins. These findings align with our previous observations of elevated body mass and increased leptin levels in both dietary unchallenged and WD-housed mutants [23,24,74]. Similar findings were obtained in both dietary challenged and unchallenged male mice lacking SERT [20] and in WD-exposed female SERT-KO rats [11]. Previous studies with KO mice have shown that that estrogen suppression is involved in SERT deficiency-induced obesity, insulin resistance, and impaired glucose tolerance, as in these mice, the aromatase (Cyp19a1) expression and levels of circulating 17β-estradiol were reduced [25]. In our study, the effect of a lack of Sert expression can be aggravated by a dietary challenge and aging. The decrease in glucose and lactate levels in KO mice suggests profound alterations in energy regulation, supported by increased IR expression in the brain, notably IRA and IRB subtypes in areas including the hypothalamus and hippocampus. These molecular changes may underlie the WD-induced relative increases in locomotion observed in KO mice, as evidenced by their performance in the open field test. This effect is consistent with ADHD-like hyperactivity in WD-exposed mice [24,33]. 

Here, it is also important to consider the broader implications of *Sert* expression and deletion in other serotonergic systems. Previous experiments have revealed altered brain expression of serotonin receptors 5-HT1A, 5-HT1B, 5-HT2A, 5-HT2C, and 5-HT6 in naïve and WD-challenged KO mice [23] which raises the question as to whether similar abnormalities might be present in peripheral organs. Specifically, *Sert* is also expressed in serotonergic neurons and the intestinal mucosa within the gut, where the latter plays a major regulatory function in the peripheral serotonergic system [76,77,78] that might contribute to the effects of Sert deficiency on metabolism. Given the known relationship between gut serotonin and various metabolic processes including glucose homeostasis and lipid metabolism, it is entirely plausible that alterations in *Sert* expression could influence these pathways. Consequently, the deletion of Sert in gut serotonergic neurons might modulate the metabolic response to a WD, potentially affecting overall metabolic health and contributing to the observed phenotypes. It is also clear that deletion of Sert in gut serotonergic neurons impacts on the microbiota. For example, we have shown that genotype had no significant impact on microbiota in animals on the Control diet, but there were significant diet-genotype interactions. After false discovery rate (FDR) correction, the WD increased Intestinimonas and Atopostipes abundance in SERT-KO animals [34].

Our metabolome study revealed decreased levels of alanine, valine, and isoleucine in KO/WD mice, aligning with findings in animal models of diabetes and rodent paradigms of WD, as well as in patients with diabetes [79,80,81,82]. These changes are also linked to clinical depression [83], supporting the idea that IR-mediated signaling plays a crucial role in depressive disorders [14,15]. 

Decreased glucose levels in WD-fed KO mice support our previous findings [23] and are in keeping with earlier reported observations in SERT-KO rats fed a high-fat, high-sucrose diet in which the animals displayed increased abdominal fat deposition with unchanged plasma glucose levels, unlike wild-type rats or male mutants [11]. Our finding contrasts with most studies where WD or high-fat diets increased blood glucose. This phenomenon may be sex-related [11,29,51] as most studies employing a high-fat, high-sugar diets using conventional or SERT-deficient strains have been carried out on male animals [20,35,36,84,85]. For example, Chen et al. (2012) reported a diet-induced increase in blood glucose levels in SERT-KO 6-month-old mice fed WD, but only males were studied in this work [20]. A recent study showed that in a high-fat and fructose post-natal dietary regime experiment, females, but not males, displayed significant decreases in brain *Sert* levels [70] suggesting sex differences in the relationship between serotonergic regulation and metabolism. Previously, it has been shown that in response to high-fat hypercaloric diet, male C57BL6 mice display elevated blood glucose levels without any signs of liver steatosis or inflammation, whereas opposing changes were reported for female mice [29]. Indeed, findings similar to our own results were obtained in hamsters fed a high-fat diet for 24 weeks [58], which seemed to be underpinned by the facilitated conversion of blood glucose to lipids. Furthermore, SERT-KO mice exhibited increased glucose absorption in the bowel and higher lipid accumulation in peripheral organs [86].

The decreased lactate levels in KO/WD mice, observed in various conditions including diabetes and depression, suggest deficits in these energy molecules may contribute to functional abnormalities in KO/WD mice [58,79,80,81,82,87].

In the open field test, the unchallenged KO mice showed reduced velocity and fewer transitions between central and peripheral sections of the arena, indicating increased anxiety-like behavior. However, KO-WD mice did not exhibit these changes. The increased velocity and transition frequency in WD-fed mutants suggest hyperactivity rather than reduced anxiety. Interestingly, grooming activity and time spent in stretched posture were increased in KO control mice and decreased in WT mice housed on WD, indicating opposing behavioral responses to dietary challenges. 

Despite unchallenged KO mice displaying hypolocomotion, significant genotype effects were found for the IRB receptor and several IR-related markers that might underlie reported behavioral changes. Of note, a relationship between ADHD and compromised IR functions is well established [40,41,42]. Exposure to WD decreased expression of isoforms of IR in the brain of KO mice in our study. Previously, lower IR expression levels in the hypothalamus were shown in a nonobese spontaneous type 2 diabetic model Goto-Kakizaki rats [88], suggesting decreased insulin signaling in development of metabolic disturbances. Hypothalamic serotonin release was reduced in Goto-Kakizaki rats in response to food intake, indicating impaired responsiveness of this neurotransmitter to food. Expression of PTEN that is tightly physiologically bound with several other proteins, indispensable for normal lipid metabolism, antioxidative defense, and anti-inflammatory response [89] was significantly altered in our study. We found changes in CD36 expression, whose functions are related to functions of PTEN. Interestingly, CD36 knockout increases whole-body insulin sensitivity (Wilson et al., 2016 [65]). 

We showed that expression of IR isoforms was paradoxically elevated in the brain of unchallenged KO mice that can be related to malfunctioning intracellular insulin signaling. Similarly, brain expression of 5-HT1B, 5-HT2A, and 5-HT2C was previously shown to be increased in the naïve KO group, whereas feeding animals with a WD paradoxically reversed these changes. This led to the hypothesis that mutants display molecular adaptations resulting in the overexpression of IR isoform, which is in keeping with clinical findings reporting that some patients with type 2 diabetes mellitus who exhibit elevated levels of tyrosine phosphatase 1B (PTP1B or PTPN1) and membrane IR-associated glycoprotein PC-1 (ENPP1), which downregulate IR phosphorylation and insulin signal transduction [67]. Specifically, PTP1B deletion beneficially affects body weight, brain insulin sensitivity, and leptin signaling in mice fed with a high-fat diet [68]. Similarly, ENPP1 overexpression stimulates severe whole-body insulin resistance, obesity, and type 2 diabetes [69]. 

Overall, our study highlights the profound impact of genetic *Sert* deficiency and WD on metabolic and behavioral profiles in mice. The elevated IR isoform expression in unchallenged KO mice suggests malfunctioning insulin signaling, supported by clinical findings of elevated PTP1B and ENPP1 in some diabetic patients [67,69]. Our results corroborate previous reports of metabolic and behavioral abnormalities in Sert-deficient mice, suggesting that further studies are warranted to understand these processes and their relevance to ADHD-like syndromes in humans.

## Figures and Tables

**Figure 1 biomolecules-14-00884-f001:**
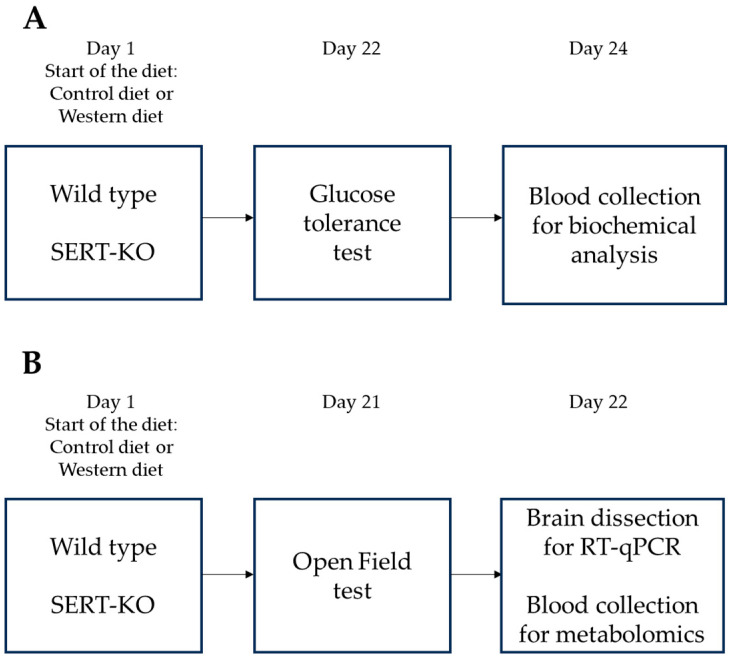
Experiment design: Flow diagram of experimental paradigm paradigm assessing the impact of the WD on WT and SERT-KO mice. (**A**) In the first study, a glucose tolerance test was carried out on Day 22, blood was harvested for biochemical assay on Day 24. (**B**) In the second study, behavioral evaluation and the open field test was performed on day 21. After the last behavioral assessment, mice were sacrificed on Day 22, and the brains were dissected, blood was collected. Brain was harvested to be used for a subsequent RNA isolation and RT-qPCR assay blood was used in the metabolome assay.

**Figure 2 biomolecules-14-00884-f002:**
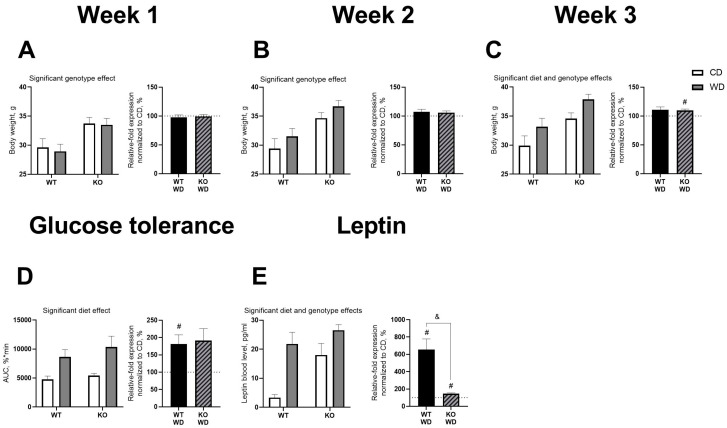
Physiological and metabolic changes in SERT-KO and WT mice on a WD. We found an effect of genotype on body weight at week 1 (**A**) and week 2 (**B**). At week 3 (**C**), significant diet and genotype effects were observed, with body weight increased in the normalized KO/WD group compared to 100%. There was an effect of diet on glucose tolerance (**D**), with the normalized WT/WD group showing an increased AUC compared to 100%. Significant diet and genotype effects on leptin levels in the blood were also found (**E**); both WT and KO mice normalized to their respective CD groups had increased leptin levels compared to 100%. Statistical analyses were performed using two-way ANOVA with Tukey’s post-hoc test, *t*-test (& *p* < 0.05), and one-sample *t*-test (# *p* < 0.05). (WT-CD group, n = 6, WT-WD group, n = 7, KO-SERT-CD group, n = 8, KO-SERT-WD group, n = 8). Black and diagonal bars represent WD-fed groups whose measurements were normalized to the respective genotype control diet group. All data are presented as mean ± SEM.

**Figure 3 biomolecules-14-00884-f003:**
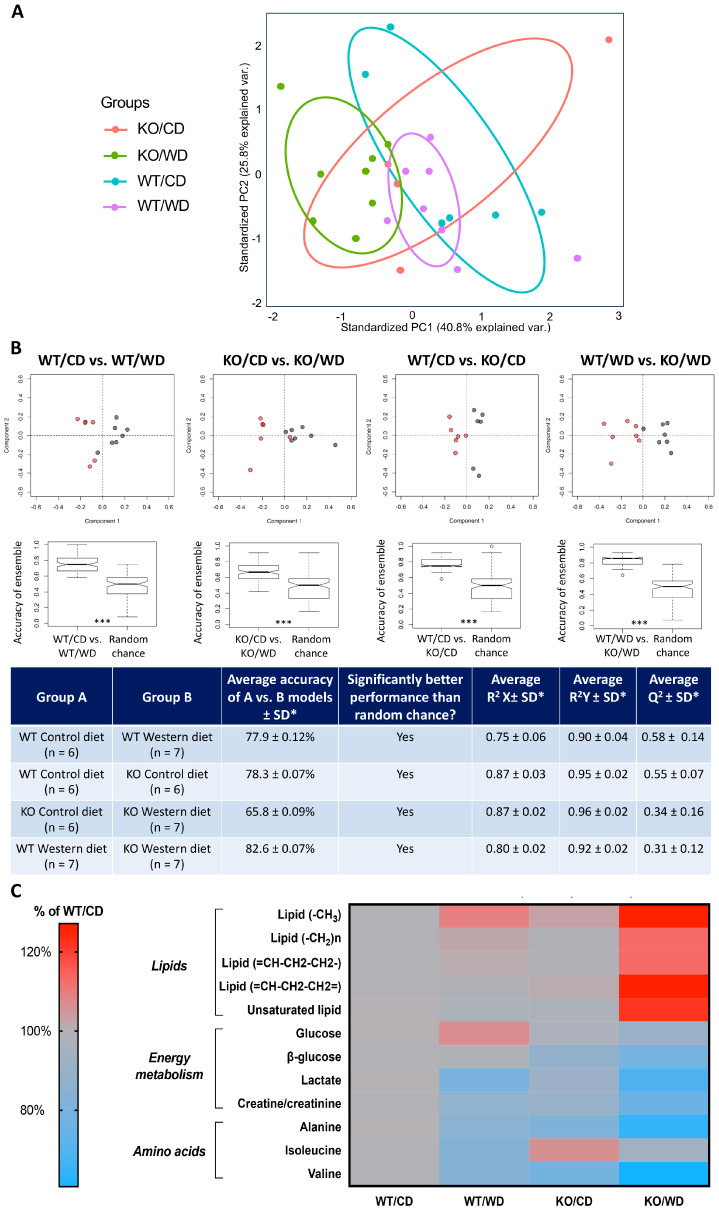
Distinct plasma metabolic profile changes in SERT-KO and WT mice housed on the WD. (**A**) PCA plot depicting how PC1 accounts for 40.8% of the variance in the data and PC2 for 25.8% which separated the data into four clusters. (**B**) OPLS-DA plots of the pairwise comparison revealed that the average accuracy for the models was high (*** *p* < 0.05, *t*-test) The true models performed significantly better than random chance in categorizing the samples in the test sets (* *p* < 0.001, Kolmogorov–Smirnov test) (**C**) A heatmap of the principal metabolites responsible for the group showing the relative relationship of each metabolite in each group. We found significant differences in the lipids, energy metabolites and amino acids in each group (n = 7 mice for each group).

**Figure 4 biomolecules-14-00884-f004:**
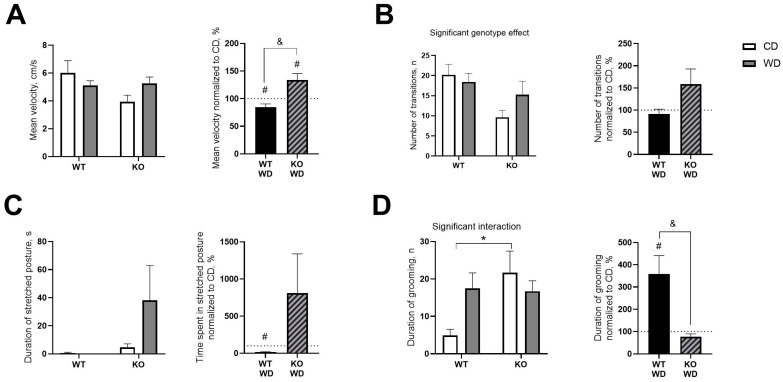
Behavioral changes in the open field between SERT-KO and WT mice fed a WD. (**A**) Velocity normalized to respective CD groups was decreased in WT/WD and increased in KO/WD mice. (**B**) There was a significant genotype effect in the number of transitions between central zone and periphery. (**C**) Time spent in stretched posture normalized to CD and duration of stretched posture was decreased in WT/WD compared to 100%. (**D**) Interaction between diet and genotype in duration of grooming was revealed, this parameter when normalized to CD was increased in WT/CD compared to 100%. Two-way ANOVA and Tukey’s post-hoc test (* *p* < 0.05), *t*-test (& *p* < 0.05) and one sample *t*-test (# *p* < 0.05). Black and diagonal bars represent WD-fed groups whose measurements were normalized to the respective genotype control diet group. (WT-CD group, n = 6, WT-WD group, n = 7, KO-SERT-CD group, n = 8, KO-SERT-WD group, n = 8). All data are mean ± SEM.

**Figure 5 biomolecules-14-00884-f005:**
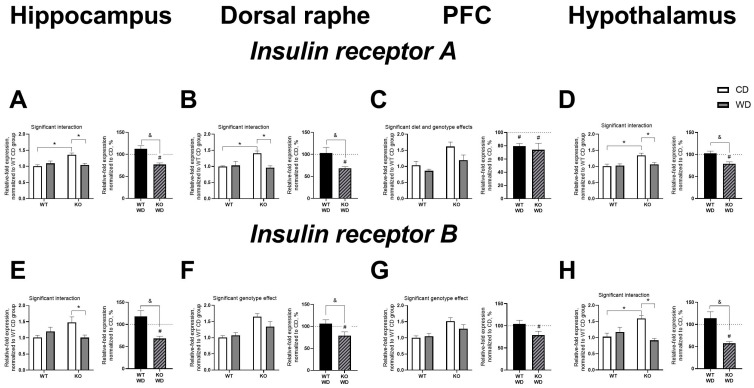
Distinct expression of IR subtypes in the brain of SERT-KO and WT mice housed on WD. IRA expression levels were increased in KO/CD compared to WT/CD mice and decreased in KO/WD compared to KO/CD in (**A**) hippocampus and (**B**) dorsal raphe. (**C**) Significant diet and genotype effects were revealed in IRA expression level in prefrontal cortex. Both WT and KO groups normalized to respective CD had this measure decreased compared to 100%. (**D**) In hypothalamus, IRA expression levels were increased in KO/CD compared to WT/CD mice and decreased in KO/WD compared to KO/CD. (**E**) IRB expression was significantly decreased in hippocampus of KO/WD compared to KO/CD. Significant genotype effect was found for IRB expression in (**F**) dorsal raphe and (**G**) prefrontal cortex. In both structures it was decreased in normalized KO mice compared to 100%. (**H**) In hypothalamus, IRB expression was significantly higher in KO/CD mice compared to WT/CD and KO/WD animals. Two-way ANOVA and Tukey’s post-hoc test (* *p* < 0.05), *t*-test (& *p* < 0.05) and one sample *t*-test (# *p* < 0.05; n = 7 mice for each group). Black and diagonal bars represent WD-fed groups whose measurements were normalized to the respective genotype control diet group. All data are mean ± SEM.

**Figure 6 biomolecules-14-00884-f006:**
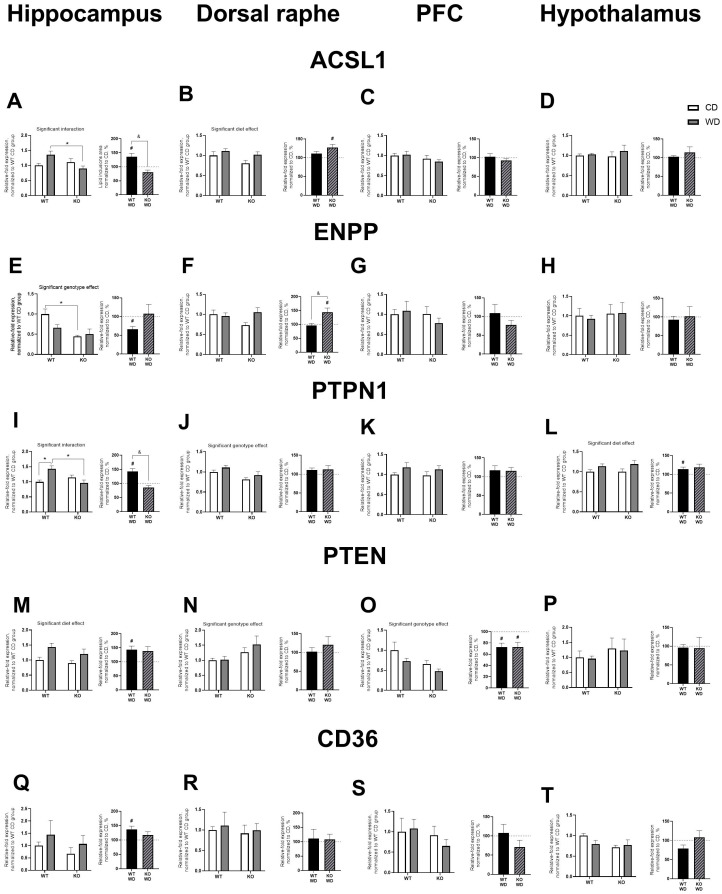
WD-induced changes in brain expression of transcription factors in SERT-KO and WT mice. ACSL1 expression in the (**A**) hippocampus, (**B**) dorsal raphe, (**C**) prefrontal cortex, and (**D**) hypothalamus. ENPP expression in the (**E**) hippocampus, (**F**) dorsal raphe, (**G**) prefrontal cortex and (**H**) hypothalamus. PTPN1 expression in the (**I**) hippocampus, (**J**) dorsal raphe, (**K**) prefrontal cortex, (**L**) hypothalamus. PTEN expression in the (**M**) hippocampus, (**N**) dorsal raphe, (**O**) prefrontal cortex, and (**P**) hypothalamus. CD36 expression in the (**Q**) hippocampus, (**R**) dorsal raphe, (**S**) prefrontal cortex, and (**T**) hypothalamus. Two-way ANOVA and Tukey’s post-hoc test (* *p* < 0.05), *t*-test (& *p* < 0.05) and one sample *t*-test (# *p* < 0.05; n = 7 mice for each group). Black and diagonal bars represent WD-fed groups whose measurements were normalized to the respective genotype control diet group. All data are mean ± SEM.

## Data Availability

Data are available upon request.

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
