# Peer review of "Impact of Serotonin Transporter Absence on Brain Insulin Receptor Expression, Plasma Metabolome Changes, and ADHD-like Behavior in Mice fed a Western Diet"

_biomolecules, 2024, doi:10.3390/biom14080884_

Round 1
Reviewer 1 Report
Comments and Suggestions for Authors
Comments and Suggestions for Authors
First, I would thank the Editorial Board of Biomolecules for allowing me to review this manuscript. The study is interesting (but complex), and the quality of English language is very good. Nonetheless, the Authors did not respect the Instructions for Authors for references, as “references must be numbered in order of appearance in the text”, “reference numbers should be placed in square brackets [ ]”, and “listed individually at the end of the manuscript” in order of appearance, and not alphabetical!
I have three major comments.
1. In my opinion the research design could be improved: why didn’t the Authors also investigate the concentrations of brain and circulating serotonin (5-HT), 5-HT receptors 5-HT1A, 5-HT2A, 5-HT3, etc., and circulating levels of estrogen, given their known implication in metabolic syndrome? These shortcomings are also present in previous studies of Authors.
2. A non-parametric statistical analysis would have been preferable, as it was more reliable, given the low number of samples, even if the data were normally distributed.
3. The discussions paragraph should be revised a little (see below).
Minor comments
INTRODUCTION
Line 57, the reference year is wrong, change it to “2011”.
Line 65, why is reported “Üçeyler et al., 2010a”, given that there is only one work by this author in the reference paragraph?
Line 70, is reported “Veniaminova et al., 2016”; is this another oversight or did Authors forget to include this article in the references paragraph?
MATERIALS AND METHODS
Lines 151-153, I would indicate the total number of mice used [twenty-six (26)?].
Lines 151-152, why did the Authors only use female mice and not male ones? Is there a specific reason? If “yes”, which one?
Line 159, please, enter the authorization number of the local ethics Committee.
Line 185, delete “per group”.
DISCUSSION
Lines 545-579, in this part of the discussions there are some statements that I don’t agree with 100%., for example:
Lines 545-546, the sentence is incomplete, “intestinal mucosa (enterochromaffin cells)” should be inserted after “Sert is also expressed in”, due to the greater importance of these intestinal cells in the regulation of the peripheral serotonergic system, compared to enteric neurons (please, see DOI: 10.1152/ajpcell.00477.2019, DOI: 10.1152/ajpgi.90685.2008, DOI: 10.1016/j.autneu.2009.08.002).
Lines 547-549, it is the liver that mainly maintains homeostasis of glucose, lipids and amino acids. Since SERT is not present in the hepatocytes of adult mice (please see link: https://www.ncbi.nlm.nih.gov/gene/15567), the liver should function the same in WT and KO mice.
Line 555, please, insert “(false discovery rate)” after “FDR”.
Lines 562-564, how do the Authors explain this contrast?
Lines 578-579, the reference given is not suitable! In the cited article, the Author himself (Altfas, J.R.) declares that: “generalizability of conclusions based on the data in this report are limited by a modest sample size, and uncertainty that the bariatric patients in the sample were truly representative of the general obese population. Results could be misleading, etc.”. Please, rephrase this sentence, or change the reference.
REFERENCES
References #8,#11,#12,#18,#21,#32,#42,#50,#53,#60,#62, there are too many Authors’ names; please, after the tenth Author name insert “et al.”, in accordance with the Instructions for Authors of “Biomolecules”.
FIGURES AND CAPTIONS
Figure 1A,B, day numbers don’t correspond to those reported in the figure 1 legend.
Figure 4, “CD” and “WD” colored boxes are missing.
Figure 4 caption (line 401), change “Velocty” with “Velocity”.
TABLES
Table S1, the more correct abbreviation of “grams” is “g” and not “gm”.

Author Response
1.First, I would thank the Editorial Board of Biomolecules for allowing me to review this manuscript. The study is interesting (but complex), and the quality of English language is very good. Nonetheless, the Authors did not respect the Instructions for Authors for references, as “references must be numbered in order of appearance in the text”, “reference numbers should be placed in square brackets [ ]”, and “listed individually at the end of the manuscript” in order of appearance, and not alphabetical!
We are grateful to Reviewer 1 for their positive view of our work and for their valuable comments and criticisms. We apologize for not complying with the Instructions for Authors regarding references. We have now ensured that all references are numbered in the order of appearance in the text, and the reference numbers are placed in square brackets.
- In my opinion the research design could be improved: why didn’t the Authors also investigate the concentrations of brain and circulating serotonin (5-HT), 5-HT receptors 5-HT1A, 5-HT2A, 5-HT3, etc., and circulating levels of estrogen, given their known implication in metabolic syndrome? These shortcomings are also present in previous studies of Authors.
We thank Reviewer 1 for the valuable suggestion regarding the research design improvement. We fully agree that it is highly relevant to study 5-HT receptors 5-HT1A, 5-HT2A, 5-HT3, and possibly other 5-HT receptors under the experimental conditions employed here. Previously, we described gene expression changes of 5-HT receptors 5-HT1A, 5-HT1B, 5-HT2A, 5-HT2C, and 5-HT6 in the prefrontal cortex, hypothalamus, dorsal raphe region, and hippocampus of aged SERT-KO mice housed on a Western diet for three weeks (Veniaminova et al., 2020a).
To recap, the brain expression of 5-HT1B, 5-HT2A, and 5-HT2C was increased in the KO group fed with CD, whereas feeding with WD paradoxically reversed these changes. SERT-KO mice, irrespective of diet, demonstrated decreased 5-HT1A and 5-HT6 expression in the dorsal raphe region.
Specifically, we found that among WD-challenged groups, 5-HT1A expression in the dorsal raphe region was significantly decreased in SERT-KO but not in wild-type mice. In the prefrontal cortex, 5-HT1A gene expression was reduced in WD-fed mice of both genotypes. In the hippocampus, there was a significant interaction between genotype and diet in levels of 5-HT2A expression, which was elevated in non-manipulated SERT-KO mice and unchanged in mutants housed on WD. A significant interaction between genotype and diet in the expression of the 5-HT1B receptor was found in the hypothalamus, dorsal raphe region, and prefrontal cortex, which was elevated in these brain regions of non-challenged mutants but not in dietary-challenged SERT-KO mice. Additionally, 5-HT1B expression was decreased in the hypothalamus and dorsal raphe region of the SERT-KO mice fed with WD compared to the non-challenged mutants.
We found a significant interaction between genotype and diet in the levels of expression of the 5-HT2C receptor in the hypothalamus, dorsal raphe region, and prefrontal cortex, with elevated expression levels in naïve mutants compared to respective controls. WD-fed SERT-KO mice revealed decreased expression of the 5-HT2C receptor in the hypothalamus and dorsal raphe region compared to naïve mutants.
There was a significant effect of genotype on the 5-HT6 receptor in the hippocampus and dorsal raphe region, where in the latter, this measure was decreased in non-manipulated but not in challenged mutants.
These changes in 5-HT regulation under basal and challenged conditions may contribute to the altered behavioral and metabolic responses of SERT-KO mice described in the present paper. In the revised manuscript version, we have included these findings in the Discussion section:
“Previous experiments have revealed altered brain expression of serotonin receptors 5-HT1A, 5-HT1B, 5-HT2A, 5-HT2C and 5-HT6 in naïve and WD-challenged KO mice (Veniaminova et al., 2020a) which raises the question as to whether similar abnormalities might be present in peripheral organs. Sert is expressed in serotonergic neurons and the intestinal mucosa within the gut, where the latter plays a major regulatory function in the peripheral serotonergic system (Bischof et al., 2009; Bertrand and Bertrand, 2010; Holton et al., 2020) that might contribute to the effects of Sert deficiency on metabolism. Given the known relationship between gut serotonin and various metabolic processes, including glucose homeostasis and lipid metabolism, it is plausible that alterations in Sert expression could influence these pathways.”
and
“We showed that expression of IR isoforms was paradoxically elevated in the brain of unchallenged KO mice that can be related to malfunctioning intracellular insulin signaling. Similarly, brain expression of 5-HT1B, 5-HT2A, and 5-HT2C was previously shown to be increased in the naïve KO group, whereas feeding animals with a WD paradoxically reversed these changes. This led to hypothesis that mutants display molecular adaptations resulting in the overexpression of IR isoform, which is in keeping with clinical findings reporting that some patients with type 2 diabetes mellitus who exhibit elevated levels of tyrosine phosphatase 1B (PTP1B or PTPN1) and membrane IR-associated glycoprotein PC-1 (ENPP1), which downregulate IR phosphorylation and insulin signal transduction (Catalano et al., 2014)”.
As suggested by Reviewer 1, the measurement of the concentration of serotonin (5-HT) would be an important add in our future studies, given that, on one hand, 5-HT brain levels are not altered in SERT-ko mice and rats (Murphy et al., 2004; Kalueff et al., 2007), and on another hand, prolonged exposure to a high fat high sure diet was shown to suppress SERT levels (Koopman et al., 2013). We also agree that the expression of 5-HT3 receptor would be of high interest to be investigated, perhaps, this is particular important with regards to gut 5-HT3 receptor, given its suggested role in increased gut leaking in SERT-deficient animals (Pomytkin et al., 2015).
We hope that in the future experiments it will be possible to address all missing elements of 5-HT system under the conditions employed here. However, in the current work we had to compromise on the molecular outcomes that we were able to employ, as the gene expression analysis undertaken utilized all available tissue.
Finally, we thank this Reviewer for a recommendation of measuring the circulating levels of estrogen given their known implication in metabolic syndrome in SERT-KO female mice housed on WD (Zha et al., 2017), while data in humans do not support a relationship between SERT and estradiol (Bethea et al., 2015). We plan to perform estrogen measurement in SERT-KO mice housed on WD in the near future that was, unfortunately, not possible in the current study, as the entire serum volume had to be used for a metabolome assay, a primary goal of our work. This aspect is discussed in the paper now, new reference is added:
“Previous studies with KO mice have shown that that estrogen suppression is involved in SERT deficiency-induced obesity, insulin resistance and impaired glucose tolerance, as in these mice, the aromatase (Cyp19a1) expression and levels of circulating 17β-estradiol were reduced (Zha et al., 2017). In our study, the effect of a lack of Sert expression can be aggravated by a dietary challenge and aging.”.
- A non-parametric statistical analysis would have been preferable, as it was more reliable, given the low number of samples, even if the data were normally distributed.
We thank Reviewer 1 for raising this important question. We have involved a professional statistician to address this issue. It is indeed true that the choice of statistical methods is critical, especially when dealing with limited sample sizes. While non-parametric methods are less sensitive to small sample sizes and are not reliant on the assumption of normal distribution, they also come with their own limitations, particularly in terms of statistical power and the ability to detect interactions between factors.
In the biomedical literature, various statistical approaches are employed in cases of limited group sizes with normally distributed data. We absolutely agree with Reviewer 1 that small sample sizes increase variability and the chance of error in statistical analysis and data interpretation. However, formally, data sets that are normally distributed are generally treated with parametric analysis.
Previous studies involving SERT-KO mice have consistently used two-way ANOVA based on the recommendations of a statistician. This method allows for the evaluation of the main effects of each factor (genotype, diet) and their interaction, which is crucial for our work. Non-parametric methods, while robust in certain contexts, do not easily facilitate the analysis of interactions between multiple factors.
For the sake of conformity with preceding work and to ensure comparability with already published results, we chose to use parametric analysis in our current manuscript. We believe that this approach provides a rigorous and consistent framework for our analysis, aligning with established methodologies in the field.
We hope that Reviewer 1 can appreciate these careful considerations and the rationale behind our choice of statistical methods.
Thank you for your understanding.
- The discussions paragraph should be revised a little (see below).
In response to the criticism expressed, the Discussion part is revised now.
- Line 57, the reference year is wrong, change it to “2011”.
Thank you for this comment, in fact, according to PubMed, full reference reads:
“Stuart, M.J., Baune, B.T., 2012. Depression and type 2 diabetes: inflammatory mechanisms of a psychoneuroendocrine co-morbidity. Neurosci Biobehav Rev. 36, 658-76. https://doi: 10.1016/j.neubiorev.2011.10.001”,
unless Reviewer 1 meant a reference in line 58:
“Giannaccini, G., Betti, L., Palego, L., Pirone, A., Schmid, L., Lanza, M., Fabbrini, L., Pelosini, C., Maffei, M., Santini, F. et al., 2011. Serotonin transporter (SERT) and translocator protein (TSPO) expression in the obese ob/ob mouse. BMC Neurosci. 12, 18. https://doi: 10.1186/1471-2202-12-18”.
In fact we aimed to refer to a human study:
“Giannaccini, G., Betti, L., Palego, L., Marsili, A., Santini, F., Pelosini, C., Fabbrini, L., Schmid, L., Giusti, L., Maffei et al., 2013. The expression of platelet serotonin transporter (SERT) in human obesity. BMC neuroscience 14, 128. https://doi.org/10.1186/1471-2202-14-128”
- Line 65, why is reported “Üçeyler et al., 2010a”, given that there is only one work by this author in the reference paragraph?
We are grateful to this Reviewer for bringing this type to our attention that is now corrected.
- Line 70, is reported “Veniaminova et al., 2016”; is this another oversight or did Authors forget to include this article in the references paragraph?
Thank you, the reference “Veniaminova, E., Cespuglio, R., Markova, N., Mortimer, N., Wai Cheung, C., Steinbusch, H.W., et al., 2016. Behavioral features of mice fed with a cholesterol-enriched diet: deficient novelty exploration and unaltered aggressive behavior. Transl. Neurosci. Clin. 2, 87–95. https://doi: 10.18679/CN11-6030/R.2016.014”
is now added to a reference list.
- Lines 151-153, I would indicate the total number of mice used [twenty-six (26)?].
As recommended, the total number of animals used in the study is included in the Materials and Methods section now:
“6-8 mice per group were used in each experiment. In the first experiment, 28 mice were used (7 mice in each group, Fig.1A), in the second cohort, 31 mice were used (Fig.1B), i.e. totally, 59 mice.
9.Lines 151-152, why did the Authors only use female mice and not male ones? Is there a specific reason? If “yes”, which one?
We thank Reviewer 1 for raising this important question. Indeed, we used female mice for several reasons. Firstly, the original work describing the model of WD employed here reported that female C57Bl6 mice are significantly more susceptible to this dietary challenge compared to male mice (Comhair et al., 2011). Females demonstrated substantial liver steatosis and over-expression of molecular markers of inflammation, as well as macrophage and neutrophil infiltration in the liver, which were not found in males (Comhair et al., 2011).
Other studies addressing the role of gender in responses to high-fat/high-calorie diets have also reported significant sex-related differences (Homberg et al., 2010; Ye et al., 2024). Homberg et al. (2010) found that in a model of high-fat, high-sucrose-choice diet, female SERT-KO rats, but not males, exhibited a strong increase (54%) in abdominal fat, while no increases in plasma glucose and insulin concentrations were observed. More recently, Ye et al. (2024) showed that only female offspring fed a high-fat, high-fructose diet displayed a significant decrease in brain serotonin levels, which was not observed in males.
These findings, together with clinical studies, suggest that the serotonergic system's susceptibility to the effects of diet is greater in females. Therefore, we chose to use female mice in our work to maximize the observed effects of WD and its potential interaction with the genetic absence of Sert. Additionally, the WD paradigm has been extensively validated on female C57BL6 mice in our previous experiments (Strekalova et al., 2015, 2016; Veniaminova et al., 2016, 2017, 2020a,b).
We have added an explanation of a choice of female mice in our study, in the Introduction section:
“We choose to use female mice as female rodents are documented to be more susceptible to the effects of the WD than males (Homberg et al., 2010; Comhair et al., 2011; Ye et al., 2024) and our previous experiments extensively validated the WD paradigm in female mice (Strekalova et al., 2015, 2016; Veniaminova et al., 2017, 2020a,b).”
- Line 159, please, enter the authorization number of the local ethics Committee.
This information has been included as requested.
- Line 185, delete “per group”.
Thank you, this has been corrected.
- Lines 545-579, in this part of the discussions there are some statements that I don’t agree with 100%., for example:
Lines 545-546, the sentence is incomplete, “intestinal mucosa (enterochromaffin cells)” should be inserted after “Sert is also expressed in”, due to the greater importance of these intestinal cells in the regulation of the peripheral serotonergic system, compared to enteric neurons (please, see DOI: 10.1152/ajpcell.00477.2019, DOI: 10.1152/ajpgi.90685.2008, DOI: 10.1016/j.autneu.2009.08.002).
We like to thank Reviewer 1 for bringing to our attention this interesting literature and for their suggestion that we revise our text describing the role of Sert in the gut. Following this recommendation, this literature is included in the reference list and mentioned in the paper, the sentence on lines 545-546 is revised:
“Specifically, Sert is also expressed in serotonergic neurons and the intestinal mucosa within the gut, where the latter plays a major regulatory function in the peripheral serotonergic system (Bischof et al., 2009; Bertrand and Bertrand, 2010; Holton et al., 2020 that might contribute to the effects of Sert deficiency on metabolism.”.
- Lines 547-549, it is the liver that mainly maintains homeostasis of glucose, lipids and amino acids. Since SERT is not present in the hepatocytes of adult mice (please see link: https://www.ncbi.nlm.nih.gov/gene/15567), the liver should function the same in WT and KO mice.
Following the remark of Reviewer 1, we have revised a sentence about the role of gut serotonin in the metabolism to highlight indirect their relationship, a sentence:
“Given the known role of gut serotonin in regulating various metabolic processes, including glucose homeostasis and lipid metabolism, it is plausible that alterations in Sert expression could influence these pathways.”
was replaced with a sentence:
“Given the known relationship between gut serotonin and various metabolic processes, including glucose homeostasis and lipid metabolism, it is entirely plausible that alterations in Sert expression could influence these pathways. Consequently, the deletion of Sert in gut serotonergic neurons might modulate the metabolic response to a WD, potentially affecting overall metabolic health and contributing to the observed phenotypes.”
We hope that this structure is considered to be an improvement. On a separate note, it is important to note that our latest findings demonstrate clear differences in hepatic responses and functions between KO and WT mice, even though SERT is not expressed in hepatocytes. These changes are believed to be owing to distinct inflammatory and other processes that are currently under investigation in our labs.
- Line 555, please, insert “(false discovery rate)” after “FDR”.
This has been corrected - thank you.
- Lines 562-564, how do the Authors explain this contrast?
At the end of the 3-week dietary challenge, SERT-KO mice displayed a significant decrease in fasting blood glucose levels compared to other genotypes. This finding contrasts with most studies where Western diets (WD) or high-fat diets increase blood glucose levels. This discrepancy may be due to a sex-related bias (Comhair et al., 2011; Ye et al., 2024), as most studies involving high-fat, high-sugar diets, including those with SERT-deficient mice, were conducted on male animals (Silva et al., 2021; Chen et al., 2012; Spadaro et al., 2015; Sun et al., 2018; Hersey et al., 2021).
For instance, previous work with WD-fed SERT-KO mice, which reported a diet-induced increase in blood glucose levels, used 6-month-old male animals (Chen et al., 2012). Additionally, it has been shown that in response to a high-fat, hypercaloric diet, male C57BL6 mice exhibit elevated blood glucose levels without major signs of liver steatosis and inflammation, whereas female mice of this strain display the opposite changes (Comhair et al., 2011). Similar findings were observed after a 24-week exposure of hamsters to a high-fat diet, suggesting a facilitated conversion of blood glucose to lipids (Guo et al., 2016).
Furthermore, SERT-KO mice have been shown to exhibit increased glucose absorption in the bowel (Greig et al., 2017). These factors together may explain the observed decrease in fasting blood glucose levels in female SERT-KO mice under the experimental conditions employed in our study.
We have extended a discussion of this finding in the Discussion now, new references are added in the Reference list:
“Decreased glucose levels in WD-fed KO mice support our previous findings (Veniaminova et al., 2020a) and are in keeping with earlier reported observations in SERT-KO rats fed a high-fat, high-sucrose diet in which the animals displayed increased abdominal fat deposition with unchanged plasma glucose levels - unlike wild-type rats or male mutants (Homberg et al., 2010). Our finding contrasts with most studies where WD or high-fat diets increased blood glucose. This phenomenon may be sex-related (Homberg et al., 2010; Comhair et al., 2011; Ye et al., 2024) as most studies employing a high fat, high sugar diets using conventional or SERT-deficient strains have been carried out on male animals (Silva et al., 2021; Chen et al., 2012; Spadaro et al., 2015; Sun et al., 2018; Hersey al., 2021). For example, Chen et al (2012) reported diet-induced increase of blood glucose levels in SERT-KO 6-month-old mice fed WD, but only males were studied in this work (Chen et al., 2012). A recent study showed that in a high fat and fructose postnatal dietary regime experiment, females, but not males, displayed significant decreases in brain Sert levels (Ye et al., 2024) suggesting sex differences in the relationship between serotonergic regulation and metabolism. Previously, it has been shown that in response to high fat hypercaloric diet, male C57BL6 mice display elevated blood glucose levels without any signs of liver steatosis or inflammation, whereas opposing changes were reported for female mice (Comhair et al., 2011). Indeed, findings similar to our own results were obtained in hamsters fed a high-fat diet for 24-weeks (Guo et al., 2016), which seemed to be underpinned by the facilitated conversion of blood glucose to lipids. Furthermore, SERT-KO mice exhibited increased glucose absorption in the bowel and higher lipid accumulation in peripheral organs (Greig et al., 2017).”.
- Lines 578-579, the reference given is not suitable! In the cited article, the Author himself (Altfas, J.R.) declares that: “generalizability of conclusions based on the data in this report are limited by a modest sample size, and uncertainty that the bariatric patients in the sample were truly representative of the general obese population. Results could be misleading, etc.”. Please, rephrase this sentence, or change the reference.
In response to the comment of Reviewer 1, we replaced the above mentioned reference concerning the relationship between ADHD and the risk of obesity with new citations that, hopefully, is viewed as being more suitable: Morandini et al., 2022 (doi: 10.1016/j.clnesp.2022.10.004. PMID: 36513489 DOI: 10.1016/j.clnesp.2022.10.004); French et al., 2024 (doi: 10.3389 /fpsyt. 2024. 1343314. PMID: 38840946 PMCID: PMC11151783), Zhu et al, 2024 (Child Obes . 2024 Mar;20(2):119-127. doi: 10.1089/chi.2022.0230, PMID: 36952326).
17.References #8,#11,#12,#18,#21,#32,#42,#50,#53,#60,#62, there are too many Authors’ names; please, after the tenth Author name insert “et al.”, in accordance with the Instructions for Authors of “Biomolecules”.
We thank reviewer 1 for bringing this technical inaccuracy to our attention that has now been carefully checked and corrected.
- Figure 1A,B, day numbers don’t correspond to those reported in the figure 1 legend.
We The numbers in the figure legend has been corrected.
- Figure 4, “CD” and “WD” colored boxes are missing.
Thank you, this Figure has been corrected.
- Figure 4 caption (line 401), change “Velocty” with “Velocity”.
This typographical error has been corrected.
- Table S1, the more correct abbreviation of “grams” is “g” and not “gm”.
We have corrected this typo.
References
Veniaminova, E., Cespuglio, R., Chernukha, I., Schmitt-Boehrer, A.G., Morozov, S., Kalueff, A. V., Kuznetsova, O., Anthony, D.C., Lesch, K.-P., Strekalova, T., 2020a. Metabolic, Molecular, and Behavioral Effects of Western Diet in Serotonin Transporter-Deficient Mice: Rescue by Heterozygosity? Front. Neurosci. 14. https://doi.org/10.3389/fnins.2020.00024
Bischoff, S.C., Mailer, R., Pabst, O., Weier, G., Sedlik, W., Li, Z., Chen, J.J., Murphy, D.L., Gershon, M.D., 2009. Role of serotonin in intestinal inflammation: knockout of serotonin reuptake transporter exacerbates 2,4,6-trinitrobenzene sulfonic acid colitis in mice. American journal of physiology. Gastrointestinal and liver physiology 296, 685–695. https://doi.org/10.1152/ajpgi.90685.2008
Bertrand, P.P., Bertrand, R.L., 2010. Serotonin release and uptake in the gastrointestinal tract. Autonomic neuroscience : basic & clinical 153, 47–57. https://doi.org/10.1016/j.autneu.2009.08.002
Holton, N.W., Singhal, M., Kumar, A., Ticho, A.L., Manzella, C.R., Malhotra, P., Jarava, D., Saksena, S., Dudeja, P.K., Alrefai, W.A. et al., 2020. Hepatocyte nuclear factor-4α regulates expression of the serotonin transporter in intestinal epithelial cells. American journal of physiology. Cell physiology 318, 1294–1304. https://doi.org/10.1152/ajpcell.00477.2019
Catalano, K.J., Maddux, B.A., Szary, J., Youngren, J.F., Goldfine, I.D., Schaufele, F., 2014. Insulin resistance induced by hyperinsulinemia coincides with a persistent alteration at the insulin receptor tyrosine kinase domain. PloS one 9, e108693. https://doi.org/10.1371/journal.pone.0108693
Murphy, D.L., Lerner, A., Rudnick, G., Lesch, K.P., 2004. Serotonin transporter: gene, genetic disorders, and pharmacogenetics. Molecular interventions 4, 109–123. https://doi.org/10.1124/mi.4.2.8
Kalueff, A.V., Fox, M.A., Gallagher, P.S., Murphy, D.L., 2007. Hypolocomotion, anxiety and serotonin syndrome-like behavior contribute to the complex phenotype of serotonin transporter knockout mice. Genes, brain, and behavior 6, 389–400. https://doi.org/10.1111/j.1601-183X.2006.00270.x
Koopman, K.E., Booij, J., Fliers, E., Serlie, M.J., la Fleur, S.E., 2013. Diet-induced changes in the Lean Brain: Hypercaloric high-fat-high-sugar snacking decreases serotonin transporters in the human hypothalamic region. Molecular metabolism 2, 417–422. https://doi.org/10.1016/j.molmet.2013.07.006
Pomytkin, I.A., Cline, B.H., Anthony, D.C., Steinbusch, H.W., Lesch, K.-P., Strekalova, T., 2015. Endotoxaemia resulting from decreased serotonin tranporter (5-HTT) function: a reciprocal risk factor for depression and insulin resistance? Behav. Brain Res. 276, 111–7. https://doi.org/10.1016/j.bbr.2014.04.049
Zha, W., Ho, H.T.B., Hu, T., Hebert, M.F., Wang, J., 2017. Serotonin transporter deficiency drives estrogen-dependent obesity and glucose intolerance. Sci. Rep. 7, 1137. https://doi.org/10.1038/s41598-017-01291-5
Bethea, C.L., Reddy, A.P., Flowers, M., Shapiro, R.A., Colman, R.J., Abbott, D.H., Levine, J.E., 2015. High fat diet decreases beneficial effects of estrogen on serotonin-related gene expression in marmosets. Progress in neuro-psychopharmacology & biological psychiatry 58, 71–80. https://doi.org/10.1016/j.pnpbp.2014.11.008
Stuart, M.J., Baune, B.T., 2012. Depression and type 2 diabetes: inflammatory mechanisms of a psychoneuroendocrine co-morbidity. Neuroscience and biobehavioral reviews 36, 658–676. https://doi.org/10.1016/j.neubiorev.2011.10.001
Giannaccini, G., Betti, L., Palego, L., Marsili, A., Santini, F., Pelosini, C., Fabbrini, L., Schmid, L., Giusti, L., Maffei et al., 2013. The expression of platelet serotonin transporter (SERT) in human obesity. BMC neuroscience 14, 128. https://doi.org/10.1186/1471-2202-14-128
Veniaminova, E., Cespuglio, R., Markova, N., Mortimer, N., Wai Cheung, C., Steinbusch, H.W., Lesch, K.P., Strekalova, T., 2016. Behavioral features of mice fed with a cholesterol-enriched diet: deficient novelty exploration and unaltered aggressive behavior. Transl. Neurosci. Clin. 2 87–95. doi: 10.18679/CN11-6030/R.2016.014
Comhair, T.M., Garcia Caraballo, S.C., Dejong, C.H., Lamers, W.H., Köhler, S.E., 2011. Dietary cholesterol, female gender and n-3 fatty acid deficiency are more important factors in the development of non-alcoholic fatty liver disease than the saturation index of the fat. Nutrition & metabolism 8, 4. https://doi.org/10.1186/1743-7075-8-4
Homberg, J.R., la Fleur, S.E., Cuppen, E., 2010, Serotonin transporter deficiency increases abdominal fat in female, but not male rats Obesity (Silver Spring) 18, 137-45. https://doi:10.1038/oby.2009.139
Ye, X., Ghosh, S., Shin, B.C., Ganguly, A., Maggiotto, L., Jacobs, J.P., Devaskar, S.U., 2024. Brain serotonin and serotonin transporter expression in male and female postnatal rat offspring in response to perturbed early life dietary exposures. Frontiers in neuroscience 18, 1363094. https://doi.org/10.3389/fnins.2024.1363094
Strekalova, T., Evans, M., Costa-Nunes, J., Bachurin, S., Yeritsyan, N., Couch, Y., Steinbusch, H.M., Eleonore Köhler, S., Lesch, K.P., Anthony, D.C., 2015. Tlr4 upregulation in the brain accompanies depression- and anxiety-like behaviors induced by a high-cholesterol diet. Brain, behavior, and immunity 48, 42–47. https://doi.org/10.1016/j.bbi.2015.02.015
Strekalova, T., Costa-Nunes, J.P., Veniaminova, E., Kubatiev, A., Lesch, K.P., Chekhonin, V.P., Evans, M.C., Steinbusch, H.W., 2016. Insulin receptor sensitizer, dicholine succinate, prevents both Toll-like receptor 4 (TLR4) upregulation and affective changes induced by a high-cholesterol diet in mice. Journal of affective disorders 196, 109–116. https://doi.org/10.1016/j.jad.2016.02.045
Veniaminova, E., Cespuglio, R., Cheung, C.W., Umriukhin, A., Markova, N., Shevtsova, E., Lesch, K.-P., Anthony, D.C., Strekalova, T., 2017. Autism-Like Behaviours and Memory Deficits Result from a Western Diet in Mice. Neural Plast. 2017, 1–14. https://doi.org/10.1155/2017/9498247
Veniaminova, E., Oplatchikova, M., Bettendorff, L., Kotenkova, E., Lysko, A., Vasilevskaya, E., Kalueff, A.V., Fedulova, L., Umriukhin, A., Lesch, K.P. et al., 2020b. Prefrontal cortex inflammation and liver pathologies accompany cognitive and motor deficits following Western diet consumption in non-obese female mice. Life sciences 241, 117163. https://doi.org/10.1016/j.lfs.2019.117163
de Andrade Silva, S.C., da Silva, A.I., Braz, G.R.F., da Silva Pedroza, A.A., de Lemos, M.D.T., Sellitti, D.F., Lagranha, C., 2021. Overfeeding during development induces temporally-dependent changes in areas controlling food intake in the brains of male Wistar rats. Life sciences 285, 119951. https://doi.org/10.1016/j.lfs.2021.119951
Chen, X., Margolis, K.J., Gershon, M.D., Schwartz, G.J., Sze, J.Y., 2012. Reduced Serotonin Reuptake Transporter (SERT) Function Causes Insulin Resistance and Hepatic Steatosis Independent of Food Intake. PLoS One 7, e32511. https://doi.org/10.1371/journal.pone.0032511
Spadaro, P.A., Naug, H.L., DU Toit, E.F., Donner, D., Colson, N.J., 2015. A refined high carbohydrate diet is associated with changes in the serotonin pathway and visceral obesity. Genetics research 97, e23. https://doi.org/10.1017/S0016672315000233
Sun, W., Guo, Y., Zhang, S., Chen, Z., Wu, K., Liu, Q., Liu, K., Wen, L., Wei, Y., Wang, B., Chen, D., 2018. Fecal Microbiota Transplantation Can Alleviate Gastrointestinal Transit in Rats with High-Fat Diet-Induced Obesity via Regulation of Serotonin Biosynthesis. BioMed research international, 8308671. https://doi.org/10.1155/2018/8308671
Hersey, M., Woodruff, J.L., Maxwell, N., Sadek, A.T., Bykalo, M.K., Bain, I., Grillo, C.A., Piroli, G.G., Hashemi, P., Reagan, L.P., 2021. High-fat diet induces neuroinflammation and reduces the serotonergic response to escitalopram in the hippocampus of obese rats. Brain, behavior, and immunity 96, 63–72. https://doi.org/10.1016/j.bbi.2021.05.010
Greig, C.J., Zhang, L., Cowles, R.A., 2017. Serotonin reuptake transporter knockout mice exhibit increased enterocyte mass and intestinal glucose absorption. J. Am. Coll. Surg. 225, S157. doi: 10.1016/j.jamcollsurg.2017.07.353
Guo, W., Jiang, C., Yang, L., Li, T., Liu, X., Jin, M., Qu, K., Chen, H., Jin, X., Liu, H. et al., 2016. Quantitative Metabolomic Profiling of Plasma, Urine, and Liver Extracts by 1H NMR Spectroscopy Characterizes Different Stages of Atherosclerosis in Hamsters. Journal of proteome research 15, 3500–3510. https://doi.org/10.1021/acs.jproteome.6b00179
Morandini, H.A.E., Watson, P.A., Barbaro, P., Rao, P., 2024. Brain iron concentration in childhood ADHD: A systematic review of neuroimaging studies. Journal of psychiatric research 173, 200–209. https://doi.org/10.1016/j.jpsychires.2024.03.035
French, B., Nalbant, G., Wright, H., Sayal, K., Daley, D., Groom, M.J., Cassidy, S., Hall, C.L., 2024. The impacts associated with having ADHD: an umbrella review. Frontiers in psychiatry 15, 1343314. https://doi.org/10.3389/fpsyt.2024.1343314
Zhu, Y., Wang, N. N., Pan, D., Wang, S., 2024. Risk of Overweight and Obesity in Children and Adolescents With Attention-Deficit/Hyperactivity Disorder: A Systematic Review and Meta-Analysis. Childhood obesity (Print) 20, 119–127. https://doi.org/10.1089/chi.2022.0230
Reviewer 2 Report
Comments and Suggestions for Authors
Manuscript Biomolecules "Absence of serotonin transporter alters brain insulin receptor expression, Western diet-induced effects in plasma metabolome profile and ADHD-like behavior in mice" by D. C. Anthony et al.
In this manuscript, the authors were interested in the interaction between serotonin transporter (SERT) inactivation and Western diet (WD) on obesity and type 2 diabetes. In wild-type (WT) and Sert KO mice (twelve-month-old) fed for three weeks on a WD or control diet, they studied locomotor behavior, blood metabolome and expression of insulin receptor A and B isoforms. They report that Sert KO mice fed on WD showed an increase in body weight and in the area under the curve (AUC) for glucose tolerance compared to the control diet (CD) with higher leptin levels. The absence of Sert gave rise to a distinct metabolic profile compared to WT animals, with KO mice fed on a WD having higher levels of lipids and lipoproteins and lower levels of glucose, lactate, alanine, valine, and isoleucine compared to other groups. They found that WD/Sert KO mice also displayed anxiety-like behavior and reduced exploratory activity in an open field test. Finally, the authors report that Sert KO mice exhibit increased brain levels of both IR A and B isoforms, with WD decreasing expression of IR in the brain of Sert KO mice. This was accompanied by an increased in the negative regulator ENPP. The authors conclude that the lack of Sert leads to a unique metabolic phenotype in aged mice, characterized by dysregulated IR-related pathways.
This report is sound and of interest for the readers. However, it should be completed to be more convincing.
1-In particular the authors should provide a table with the full statistics description of their experiments.
2-For quantification, it would be useful to complete their investigations with western blots of their important findings (e.g. IR and PTPN1 proteins)
3-The introduction and discussion about ADHD could be shortened since their behavioral findings are minor and not directly relevant to this issue.
4-Some typos should be corrected (e.g. geneotype line 330)
Author Response
Reviewer 2
In this manuscript, the authors were interested in the interaction between serotonin transporter (SERT) inactivation and Western diet (WD) on obesity and type 2 diabetes. In wild-type (WT) and Sert KO mice (twelve-month-old) fed for three weeks on a WD or control diet, they studied locomotor behavior, blood metabolome and expression of insulin receptor A and B isoforms. They report that Sert KO mice fed on WD showed an increase in body weight and in the area under the curve (AUC) for glucose tolerance compared to the control diet (CD) with higher leptin levels. The absence of Sert gave rise to a distinct metabolic profile compared to WT animals, with KO mice fed on a WD having higher levels of lipids and lipoproteins and lower levels of glucose, lactate, alanine, valine, and isoleucine compared to other groups. They found that WD/Sert KO mice also displayed anxiety-like behavior and reduced exploratory activity in an open field test. Finally, the authors report that Sert KO mice exhibit increased brain levels of both IR A and B isoforms, with WD decreasing expression of IR in the brain of Sert KO mice. This was accompanied by an increased in the negative regulator ENPP. The authors conclude that the lack of Sert leads to a unique metabolic phenotype in aged mice, characterized by dysregulated IR-related pathways.
This report is sound and of interest for the readers. However, it should be completed to be more convincing.
We are grateful to Reviewer 2 for favorable view of our work and like to thank them for their valuable comments and suggestions.
- 1. In particular the authors should provide a table with the full statistics description of their experiments.
Following this recommendation of Reviewer 2, tables with the full statistics description of the experiments have been added within a Supplementary File.
- For quantification, it would be useful to complete their investigations with western blots of their important findings (e.g. IR and PTPN1 proteins)
We fully agree with this view of Reviewer 2, and shall perform western blots of the most relevant IR-related molecular markers in the next round of our studies, as these experiments would require a complete set of new tissue to be generated.
- The introduction and discussion about ADHD could be shortened since their behavioral findings are minor and not directly relevant to this issue.
As recommended, we have revised the text in line with Reviewer 2's suggestion to tone down the discussion about ADHD in this paper. Indeed, several studies have demonstrated a link between SERT polymorphisms or SERT expression and ADHD (Murphy et al., 2004; Chatterjee et al., 2022). For example, a recent study by Kautzky et al. (2020) on a cohort of ADHD patients showed that “With a prediction accuracy above 0.8, the findings advocate the relevance of the SERT as well as the HTR1B and HTR2A genes in ADHD and hint towards disease-specific effects.” Another study reported that the inheritance of the “s” variant of the SLC6A4 gene in ADHD patients is associated with “hyperactivity and behavioral problems” (Chatterjee et al., 2022).
We originally included this topic in our manuscript to place our data in a broader clinical and preventive context. Our earlier findings in the WD model in mice led us to hypothesize the role of nutrition in the precipitation of psychiatric disorders, such as ADHD. We believe that addressing this issue in the present work could be beneficial in this respect. However, we have now reduced the length of the sections in the Introduction and Discussion that concern this link and added additional literature to explain the clinical context of our work in a more concise manner. New references have been added accordingly.
4.Some typos should be corrected (e.g. geneotype line 330)
Thank you, we have carefully checked the text to find any remaining typos.
References
Murphy, D.L., Lerner, A., Rudnick, G., Lesch, K.P., 2004. Serotonin transporter: gene, genetic disorders, and pharmacogenetics. Molecular interventions 4, 109–123. https://doi.org/10.1124/mi.4.2.8
Chatterjee, M., Saha, S., Sinha, S., Mukhopadhyay, K., 2022. A three-pronged analysis confirms the association of the serotoninergic system with attention deficit hyperactivity disorder. World journal of pediatrics : WJP 18, 825–834. https://doi.org/10.1007/s12519-022-00614-5
Kautzky, A., Vanicek, T., Philippe, C., Kranz, G.S., Wadsak, W., Mitterhauser, M., Hartmann, A., Hahn, A., Hacker, M., Rujescuet, D. et al., 2020. Machine learning classification of ADHD and HC by multimodal serotonergic data. Transl Psychiatry 10, 104. https://doi.org/10.1038/s41398-020-0781-2
Reviewer 3 Report
Comments and Suggestions for Authors
In this manuscript, the authors studied the impact of western diet and Sert deficiency on metabolic and behavioral traits. Also analyzed the different IR isoforms and the associated signaling pathways in western diet and Sert deficient female mice. Though studying the integral role of serotonin transporter and the insulin receptor is a relevant topic, unfortunately, the links between Sert and insulin signaling is poorly examined.
Major concerns
1. At any point during the experiment, have the glucose levels been checked?
2. Insulin resistance also plays a crucial role in brain health and related disorders and should have been included in the study.
3. The study only included female mice. Will there be any effect on sex hormones in this study?
4. Usually diet induced study needs long durations like 3 or 6 months, the authors used only 24 days of WD, is this animal model valid for this study? Please justify
5. Keep the introduction very brief and only write what is necessary.
6. The number of animals (n) used in the study was not mentioned by the authors in the figure legends.
7. In Figure 2, 4 & 5, the authors ought to have explained what is * and # in the legends.
8. Apart from open field test, another anxiety test like light dark test could have been done.
Author Response
Reviewer 3
In this manuscript, the authors studied the impact of western diet and Sert deficiency on metabolic and behavioral traits. Also analyzed the different IR isoforms and the associated signaling pathways in western diet and Sert deficient female mice. Though studying the integral role of serotonin transporter and the insulin receptor is a relevant topic, unfortunately, the links between Sert and insulin signaling is poorly examined.
We thank Reviewer 3 for all their recommendations.
- At any point during the experiment, have the glucose levels been checked?
Indeed, we studied glucose levels after the termination of a 3-week WD exposure using two different methods: the glucose tolerance test and metabolome analysis. The results are presented in Figures 2 and 3, respectively. In the glucose tolerance test, we did not observe significant effects of genotype on the area under the curve, although a strong effect of diet was found (Fig. 2D). In the metabolome analysis, which evaluated baseline glucose levels, a genotype x diet interaction as well as a genotype effects were found; baseline glucose levels were significantly lower in the SERT-KO/WD group compared to the WT/WD group (Fig. 3, Suppl. Fig. 2J). Our previous study also reported decreased baseline glucose levels in naïve SERT-KO mice (Veniaminova et al., 2020a). This finding is discussed in the manuscript text:
“Decreased glucose levels in WD-fed KO mice support our previous findings (Veniaminova et al., 2020a) and are in keeping with earlier reported observations in SERT-KO rats fed a high-fat, high-sucrose diet in which the animals displayed increased abdominal fat deposition with unchanged plasma glucose levels - unlike wild-type rats or male mutants (Homberg et al., 2010). Our finding contrasts with most studies where WD or high-fat diets increased blood glucose. This phenomenon may be sex-related (Homberg et al., 2010; Comhair et al., 2011; Ye et al., 2024) as most studies employing a high fat, high sugar diets using conventional or SERT-deficient strains have been carried out on male animals (Silva et al., 2021, Chen et al., 2012; Sparado et al., 2015; Sun et al., 2018; Hersey al., 2021). For example, Chen et al (2012) reported diet-induced increase of blood glucose levels in SERT-KO 6-month-old mice fed WD, but only males were studied in this work (Chen et al., 2012). A recent study showed that in a high fat and fructose postnatal dietary regime experiment, females, but not males, displayed significant decreases in brain Sert levels (Ye et al., 2024) suggesting sex differences in the relationship between serotonergic regulation and metabolism. Previously, it has been shown that in response to high fat hypercaloric diet, male C57BL6 mice display elevated blood glucose levels without any signs of liver steatosis or inflammation, whereas opposing changes were reported for female mice (Comhair et al., 2011). Indeed, findings similar to our own results were obtained in hamsters fed a high-fat diet for 24-weeks (Guo et al., 2016), which seemed to be underpinned by the facilitated conversion of blood glucose to lipids. Furthermore, SERT-KO mice exhibited increased glucose absorption in the bowel and higher lipid accumulation in peripheral organs (Greig et al., 2017).”
- Insulin resistance also plays a crucial role in brain health and related disorders and should have been included in the study.
We fully agree with this recommendation, this assay has been carried out in our previous study with female C57BL6 mice. We found that a 3-week exposure of these animals to WD has induced a marketable insulin resistance (Veniaminova et al., 2020b). For technical reasons, this assay could not be included in a study design of the present work, as it requires a separate cohort of experimental animals. Because of small volumes of serum that can be obtained from mice, it was also not possible to measure insulin blood levels, as the whole amount was used for a metabolome assay.
We hope that the investigation of insulin resistance in SERT-KO mice fed WD can be done in our future studies that would be one of our high research priorities in this line of work.
- The study only included female mice. Will there be any effect on sex hormones in this study?
The question as for our choice of sex of animals used echoes with a question of Reviewer 1 (please kindly see our response to his question 9). We chose to use female mice since female rodents are well documented to be more susceptible to the effects of the WD than males (Homberg et al., 2009; Comhair et al., 2011; Ye et al., 2024).
We agree with Reviewer 3 that the question about the levels of sex hormones in experimental groups is of considerable interest. In fact, it has been recently demonstrated that SERT-KO mice have reduced circulating 17β-estradiol levels, the replacement of which reverses the obesity and compromised glucose intolerance (Zha et al., 2017). This was shown to be owned to suppressed aromatase (Cyp19a1) expression in SERT-KO mice (Zha et al., 2017). As such, this effect of genetic lack of SERT on estrogen production, along with additive suppressive effects of aging, are likely to be implicated in reported here metabolic abnormalities including compromised glucose tolerance. We like to plan to study estrogen concentrations under employed here experimental conditions in our future experiments that was not possible to study because the study of metabolome profile required to use entire amount of serum collected from the animals. We have discussed this issue in the ms text now.
“Previous studies with KO mice demonstrated that estrogen suppression is involved in SERT deficiency-induced obesity, insulin resistance and impaired glucose tolerance, as in these mice, the aromatase (Cyp19a1) expression and levels of circulating 17β-estradiol are reduced (Zha et al., 2017). In our study, these effects of genetic lack of Sert can be aggravated by a dietary challenge and aging.“.
At the same time, we trust that estrogen cycle is unlikely to affect the outcome in our study since long co-housing of female mice in the same laboratory room was shown to result in a synchronization of the cycle. Furthermore, the use of aged mice, at a point at which they are not longer fertile and display a strong drop in sex hormone levels, is likely to limit the impact of sex hormones in the genotype differences reported in our paper.
- Usually diet induced study needs long durations like 3 or 6 months, the authors used only 24 days of WD, is this animal model valid for this study? Please justify
We agree with Reviewer 3 that various experimental protocols with different durations of WD exposure are used in rodents to induce metabolic syndrome, liver steatosis, and impairment of glucose tolerance. In this work, we employed a 3-week WD exposure paradigm in mice, first proposed by Comhair et al. (2011), which was rigorously studied by this group for peripheral WD-induced changes. This protocol was further investigated for both peripheral and CNS WD-induced changes and validated using pharmacological interventions in our laboratories (Strekalova et al., 2015, 2016; Veniaminova et al., 2017, 2020a,b). In the original study by the Comhair group, as well as in all subsequent experiments, impairment of glucose tolerance, liver steatosis, and sterile inflammation were reproducibly reported in female C57BL6 mice. Therefore, to be consistent with our previous results obtained in aged SERT-KO mice, we applied a 3-week WD challenge in the present study.
It can be hypothesized that other researchers aiming to induce these WD-induced abnormalities have exposed their experimental animals to a substantially longer high-fat/high-sugar dietary regimen due to the use of male rodents, which are much less sensitive to the WD, as discussed in Section 3 of this Response letter and Section 9 of the Response to Reviewer 1 (see above).
- Keep the introduction very brief and only write what is necessary.
We thank this Reviewer for this suggestion that we have reduced the length of the introduction where possible.
- The number of animals (n) used in the study was not mentioned by the authors in the figure legends.
We are grateful to Reviewer 3 for bringing up this technical comment, respective Figure legends are completed now with the information on the number of animals that were used.
- In Figure 2, 4 & 5, the authors ought to have explained what is * and # in the legends.
Figure legends have been revised.
- Apart from open field test, another anxiety test like light dark test could have been done.
We fully agree with Reviewer 3 that to effectively demonstrate anxiety-like changes, other tests besides the open field test should be used. We did not conduct these tests in the present work for several reasons. Firstly, these tests have already been performed in previous studies using aged SERT-KO mutants. For instance, the O-maze test was conducted on 12-month-old WT and SERT-KO mice housed on the WD for 3 weeks (Veniaminova et al., 2020a). In this test, there was a significant diet effect on the latency to exit to the open arm, which decreased in the KO mice fed with WD compared to those fed with CD. This finding could be interpreted as a sign of increased impulsivity, a feature induced by WD in mice (Strekalova et al., 2015; Veniaminova et al., 2017).
Secondly, because WD-fed mice display hyperactivity (Strekalova et al., 2015, 2016; Veniaminova et al., 2017), the use of classic behavioral tests for anxiety, which can be highly stressful for these mice, may compromise the evaluation of anxiety under these conditions. This was the case in a study of dark/light box behavior in one of our former experiments with young WD-fed mice (Strekalova et al., 2015).
References
Veniaminova, E., Cespuglio, R., Chernukha, I., Schmitt-Boehrer, A.G., Morozov, S., Kalueff, A. V., Kuznetsova, O., Anthony, D.C., Lesch, K.-P., Strekalova, T., 2020a. Metabolic, Molecular, and Behavioral Effects of Western Diet in Serotonin Transporter-Deficient Mice: Rescue by Heterozygosity? Front. Neurosci. 14. https://doi.org/10.3389/fnins.2020.00024
Comhair, T.M., Garcia Caraballo, S.C., Dejong, C.H., Lamers, W.H., Köhler, S.E., 2011. Dietary cholesterol, female gender and n-3 fatty acid deficiency are more important factors in the development of non-alcoholic fatty liver disease than the saturation index of the fat. Nutrition & metabolism 8, 4. https://doi.org/10.1186/1743-7075-8-4
Homberg, J.R., la Fleur, S.E., Cuppen, E., 2010, Serotonin transporter deficiency increases abdominal fat in female, but not male rats Obesity (Silver Spring) 18, 137-45. https://doi:10.1038/oby.2009.139
Ye, X., Ghosh, S., Shin, B.C., Ganguly, A., Maggiotto, L., Jacobs, J.P., Devaskar, S.U., 2024. Brain serotonin and serotonin transporter expression in male and female postnatal rat offspring in response to perturbed early life dietary exposures. Frontiers in neuroscience 18, 1363094. https://doi.org/10.3389/fnins.2024.1363094
de Andrade Silva, S.C., da Silva, A.I., Braz, G.R.F., da Silva Pedroza, A.A., de Lemos, M.D.T., Sellitti, D.F., Lagranha, C., 2021. Overfeeding during development induces temporally-dependent changes in areas controlling food intake in the brains of male Wistar rats. Life sciences 285, 119951. https://doi.org/10.1016/j.lfs.2021.119951
Chen, X., Margolis, K.J., Gershon, M.D., Schwartz, G.J., Sze, J.Y., 2012. Reduced Serotonin Reuptake Transporter (SERT) Function Causes Insulin Resistance and Hepatic Steatosis Independent of Food Intake. PLoS One 7, e32511. https://doi.org/10.1371/journal.pone.0032511
Spadaro, P.A., Naug, H.L., DU Toit, E.F., Donner, D., Colson, N.J., 2015. A refined high carbohydrate diet is associated with changes in the serotonin pathway and visceral obesity. Genetics research 97, e23. https://doi.org/10.1017/S0016672315000233
Sun, W., Guo, Y., Zhang, S., Chen, Z., Wu, K., Liu, Q., Liu, K., Wen, L., Wei, Y., Wang, B., Chen, D., 2018. Fecal Microbiota Transplantation Can Alleviate Gastrointestinal Transit in Rats with High-Fat Diet-Induced Obesity via Regulation of Serotonin Biosynthesis. BioMed research international, 8308671. https://doi.org/10.1155/2018/8308671
Hersey, M., Woodruff, J.L., Maxwell, N., Sadek, A.T., Bykalo, M.K., Bain, I., Grillo, C.A., Piroli, G.G., Hashemi, P., Reagan, L.P., 2021. High-fat diet induces neuroinflammation and reduces the serotonergic response to escitalopram in the hippocampus of obese rats. Brain, behavior, and immunity 96, 63–72. https://doi.org/10.1016/j.bbi.2021.05.010
Greig, C.J., Zhang, L., Cowles, R.A., 2017. Serotonin reuptake transporter knockout mice exhibit increased enterocyte mass and intestinal glucose absorption. J. Am. Coll. Surg. 225, S157. doi: 10.1016/j.jamcollsurg.2017.07.353
Guo, W., Jiang, C., Yang, L., Li, T., Liu, X., Jin, M., Qu, K., Chen, H., Jin, X., Liu, H. et al., 2016. Quantitative Metabolomic Profiling of Plasma, Urine, and Liver Extracts by 1H NMR Spectroscopy Characterizes Different Stages of Atherosclerosis in Hamsters. Journal of proteome research 15, 3500–3510. https://doi.org/10.1021/acs.jproteome.6b00179
Veniaminova, E., Oplatchikova, M., Bettendorff, L., Kotenkova, E., Lysko, A., Vasilevskaya, E., Kalueff, A.V., Fedulova, L., Umriukhin, A., Lesch, K.P. et al., 2020b. Prefrontal cortex inflammation and liver pathologies accompany cognitive and motor deficits following Western diet consumption in non-obese female mice. Life sciences 241, 117163. https://doi.org/10.1016/j.lfs.2019.117163
Zha, W., Ho, H.T.B., Hu, T., Hebert, M.F., Wang, J., 2017. Serotonin transporter deficiency drives estrogen-dependent obesity and glucose intolerance. Sci. Rep. 7, 1137. https://doi.org/10.1038/s41598-017-01291-5
Strekalova, T., Evans, M., Costa-Nunes, J., Bachurin, S., Yeritsyan, N., Couch, Y., Steinbusch, H.M., Eleonore Köhler, S., Lesch, K.P., Anthony, D.C., 2015. Tlr4 upregulation in the brain accompanies depression- and anxiety-like behaviors induced by a high-cholesterol diet. Brain, behavior, and immunity 48, 42–47. https://doi.org/10.1016/j.bbi.2015.02.015
Strekalova, T., Costa-Nunes, J.P., Veniaminova, E., Kubatiev, A., Lesch, K.P., Chekhonin, V.P., Evans, M.C., Steinbusch, H.W., 2016. Insulin receptor sensitizer, dicholine succinate, prevents both Toll-like receptor 4 (TLR4) upregulation and affective changes induced by a high-cholesterol diet in mice. Journal of affective disorders 196, 109–116. https://doi.org/10.1016/j.jad.2016.02.045
Veniaminova, E., Cespuglio, R., Cheung, C.W., Umriukhin, A., Markova, N., Shevtsova, E., Lesch, K.-P., Anthony, D.C., Strekalova, T., 2017. Autism-Like Behaviours and Memory Deficits Result from a Western Diet in Mice. Neural Plast. 2017, 1–14. https://doi.org/10.1155/2017/9498247

Round 2
Reviewer 1 Report
Comments and Suggestions for Authors
Comments and Suggestions for Authors
The Authors have given positive attention to my comments, so the manuscript has been improved.
Minor comments
Reference #1, add “T.” after the family name “Moriyama”.
Reference #13, add “M.” after the family name “Maffei”.
Reference #50, add “D.” after the family name “Leppert”.
Table S1, there are still some “gm” to change to “g”.
Reviewer 3 Report
Comments and Suggestions for Authors
The authors responded to the comments essentially in an appropriate manner. The manuscript has been improved markedly. I would appreciate the authors if they can include the insulin resistance experiment.